

# Nepal Emission Inventory (NEEMI): a high resolution technology-based bottom-up emissions inventory for Nepal 2001-2016

Pankaj Sadavarte*[a], Maheswar Rupakheti*[a], Prakash V. Bhave[b], Kiran Shakya[b], Mark G. Lawrence[a]

[a]Institute for Advanced Sustainability Studies (IASS), Berliner Str. 130, 14467 Potsdam, Germany
[b]International Centre for Integrated Mountain Development (ICIMOD), Lalitpur, Nepal

*Corresponding Authors: Pankaj Sadavarte (p.sadavarte@sron.nl) and Maheswar Rupakheti (Maheswar.rupakheti@iass-potsdam.de)



## Abstract

The lack of a comprehensive, up-to-date emission inventory for the Himalayan region is a major challenge in understanding the regional air pollution, including its impacts, mitigation, and the relevant atmospheric processes. This study develops a high resolution (1 km × 1 km) present-day emission inventory for Nepal with a higher-tier approach (detailed) to understanding the current combustion technologies and sectoral energy consumption. We estimate emissions of aerosols, trace gases and greenhouse gases from five energy-use sectors (residential, industry, commercial, agriculture and transport) and an open-burning source (agro-residue) for the period 2001–2016 (with 2011 as the base year), using bottom-up methodologies. Newly-measured country-specific emission factors (EFs) are used for emission estimates. It is estimated that the national total energy consumption in 2011 was 378 PJ with the residential sector being the largest energy consumer (79 %), followed by the industry (11 %) and transport (7 %) sectors. Biomass is the dominant energy source contributing 88 % to national total energy consumption, while the share of fossil fuel is only 12 %. With regards to open burning of the crop waste, it is estimated that 9.3 million tons of agro-waste was burned after harvesting crops in 2011. Nationally, 8.4 Tg $CO_2$, 666 Gg $CH_4$, 2.5 Gg $N_2O$, 72 Gg $NO_X$, 1984 Gg CO, 477 Gg NMVOC, 239 Gg $PM_{2.5}$, 28 Gg BC, 99 Gg OC and 28 Gg $SO_2$ were emitted from these sources in 2011. The energy consumption was also estimated for each year for the period 2001–2016 which shows an increase by a factor of 1.6 in 2016, while the emissions of various species increased by a factor of 1.2–2.4 with respect to 2001. An assessment of the top polluting technologies shows high emissions from traditional cookstoves using firewood, dungcakes, and agricultural residues, and open burning emissions of wood and residues. In addition, high emissions were also encountered from fixed chimney Bull's Trench kilns for brick production, cement kilns, two-wheeler gasoline vehicles, heavy diesel freight vehicles and kerosene lamps. A GIS-based gridded 1 km × 1 km population density map incorporating land-use and land cover data, settlement points, and topography was used for the spatial distribution of residential emissions. Geospatial locations were assigned to point sources, while activity-based proxies were used for other sources. Emissions were apportioned across different months from brick production, the agriculture sector, diesel generators, and space and water heating, using respective temporal variations of the activities. It was found that April had the maximum $PM_{2.5}$ emissions, followed by December, January and February. Also, a wide variation in emissions



distribution was found, highlighting the pockets of growing urbanization and the detailed knowledge about the emission sources. These emissions will be of value for further studies, especially air quality modelling studies focused on understanding the likely effectiveness of air pollution mitigation measures in Nepal.

**Keywords**

*Nepal emission inventory, aerosols, residential, diesel generator sets, transport, high resolution emissions, Kathmandu Valley*



## 1. Introduction

Nepal, a developing country in South Asia is subject to the same increasing burden of air pollution seen over most of the subcontinent for the last decades (MoEST, 2005). With its complex topography ranging from ca. 100 m above sea level (asl) in the south to above 8000 m asl in the north, the region is home to

a population of 26.5 million (in 2011) that depends mainly on biomass and  fossil fuel (imported from other countries) for its total energy needs (CBS, 2012; WECS, 2014). It is well established that the incomplete combustion of such biomass emits a significant amount of the fine particulates $PM_{2.5}$ (diameter ≤ 2.5 µm) and ozone precursor emissions, which have been to linked to degrading air quality, adverse health impacts, climate change and effects on the cryosphere (Fiore et al., 2015; Shakya et al.,

2016). In addition, studies have also shown air pollution causes reductions in crop productivity, and have identified hotspots and shown the air pollution linkage behind urban heat islands, altering monsoon patterns and increases in natural calamities like floods (Burney and Ramanathan, 2014; Shastri et al., 2017; Collier and Zhang, 2009; Fan et al., 2015).

Global studies of health effects have identified air pollution, including both household pollution and ambient particulate pollution, as the 2nd and 3rd leading risk factors responsible for burden of disease attributable to premature deaths in Nepal (Forouzanfar, 2016). The prolonged exposure to these pollutants have led to significant respiratory symptoms. The exposure of adults to biomass smoke in rural households of Nepal has led to prevailing respiratory symptoms, while those with prolonged exposure to

ambient particulate pollution in urban households have shown evidence of high chronic phlegm (Kurmi, 2014). A study has also demonstrated intake as high as 165 µg m$^{-3}$ respirable fraction of particulates in childrens' from the use of biomass for residential activities in Nepal (Devakumar et al., 2014). The roadside measurements in the Valley have also observed an average of ca. 90 µg m$^{-3}$ of $PM_{2.5}$ concentration with higher upper bounds during the winter season (Bhari, 2015; Shakya et al., 2017). The

ambient measurements of $PM_{2.5}$ in the Kathmandu Valley, the capital and main metropolitan region in Nepal, have shown seasonal concentrations varying from 30 µg m$^{-3}$ (monsoon) to 90 µg m$^{-3}$ (winter), attributed to various emission sources and meteorology (Aryal et al., 2009). Such large variations in ambient concentrations are also likely partly due to seasonality in energy consumption and emissions.



The GDP of Nepal has been increasing substantially in the past two decades (MoF, 2017). Similarly there has been tremendous increase in the energy-use footprint and import of fossil fuels (NOC, 2019). Since a large fraction of available national energy is consumed in the residential sector relying on biomass fuels, there is an urgent need to understand the demand and supply of bio-resources and explore other cleaner options. Moreover, the rapid urbanization has led to an increase in vehicle registration by about 15-fold in the last two decades, unfortunately increasing the demand for petroleum fuels (DoTM, 2016). The increasing population and commercialization also calls for uninterrupted electricity. Until recently, Nepal has faced very high amounts of load shedding and diesel consumption in diesel generators for backup power generation. This has led to a significant increase in the national black carbon (BC) emissions (World Bank, 2014). In order to tackle the problem of future energy demand (though the situation has improved since 2016) and degrading air quality, synergetic work is needed based on an understanding of the current fuel consumption and efficient combustion technologies.

Recent studies in Nepal have attempted to characterize and quantify the energy needs and emissions from sources such as the residential and commercial sectors using the TIER-I approach (less detailed) that fails to provide complete information about the combustion technologies and control abatements (Malla, 2013; Bhattarai and Jha, 2015). Research studies have mainly focused on residential cooking with an aim to promote energy-efficient and fuel-efficient cookstoves, renewable technologies, and indoor air quality, with a brief discussion about the co-benefits of cleaner combustion technologies (Pokhrel, 2015; Gurung et al., 2012; Singh et al., 2012). Similarly, past studies on the transport sector have only been focused on the Kathmandu Valley due to large number of vehicles registered in Bagmati zone, where the Kathmandu Valley is located, and subsequently higher sales of gasoline and diesel in the Valley (Shrestha et al., 2013). Past studies like Shrestha and Rajbhandari (2010) have modelled the influence of the residential, agricultural, transport, industrial and commercial sectors on reducing future carbon emissions only for Kathmandu Valley, which takes into consideration the economics and demands of the population without a detailed classification of combustion technologies that drive the emissions.



Moreover, current Asian emission inventories tend to provide explicit details on energy use patterns only for regions of interest, while aggregating the same on a coarser resolution for the rest of the countries (Zhang et al., 2009; Kurokawa et al., 2013). One such effort in segregating the energy use at a higher resolution of sectoral activities has been executed by the Water and Energy Commission's Secretariat (WECS), Nepal; however, there is no clear methodology and the underlying assumptions stated which can be relied on for further integration into emissions estimates (WECS, 2014). In addition to the arousing air pollution strategies, the NAMaSTE campaign has led to the measurements of emission factors from the different informal sources in Nepal, thereby providing Nepal-specific emission factors  (EFs) that can be used in reducing the uncertainties in emissions (Stockwell et al., 2016; Jayarathane et al., 2017).

Analyzing the following issues, it is important to conduct a systematic comprehensive study of all energy sectors (i.e., technology-based production and use of energy) in Nepal from an emissions point of view, which has not yet been conducted, integrating the primary information on energy production and use, fuel combustion technologies and corresponding EFs. This study has developed a high resolution 1 km × 1 km technology-based emission inventory of ten species, including greenhouse gases and short-lived climate-forcing pollutants (SLCPs). The methodology of developing emissions estimates is described in Sect. 2, followed by information on activity rates, the detailed combustion technology and industrial process in practice, the region-specific EFs, spatial surrogates, and finally sectoral emissions. The analysis of our results compares our national energy and emissions with reported estimates by national authorities and other regional studies. Finally, the emissions from the Kathmandu Valley are compared with the rest of the Nepal, to understand the relative contributions of different sources and emissions at the national level as well as in the Kathmandu Valley.

## 2.  Methodology

The methodology follows a bottom-up estimation of emissions using activity rates in the form of fuel consumption and the measured EFs in Nepal, wherever possible. The fuel consumption is uniquely estimated across each sub-sector using available primary data of activity rates such as fuel imports, production, registered number of vehicles, machinery units, and specific energy consumption (SEC). The





approach also incorporates current industrial process technologies and the penetration of control measures for particulates and other species (Fig. 1).

An emission *'E'* is estimated using Eq. (1), where pollutant *'i'* from sub-sector *'a'* is calculated using the
fuel consumed *'FC'* by its type *'f'* in combustion technology *'t'* and emission factor *'EF'*. The emission factor considered is a function of fuel type and the combustion technology, different across each sub-sector and fuel-technology combination. The black carbon (BC) and organic carbon (OC), two key components of particulate matter, are calculated using their respective fractions of $PM_{2.5}$ (in Sect. 2.3) (Eq. 2 & 3). Table 1 lists the complete details about the emission sectors, pollutant species and spatial
resolution considered in this study, referred to as NEpal EMission Inventory (NEEMI).

$$E_{i,a} = FC_{f,a,t} \times EF_{i,f,a,t} \tag{1}$$

$$E_{BC,a,f,t} = E_{PM2.5,a,f,t} \times f_{BC,a,f,t} \tag{2}$$

$$E_{OC,a,f,t} = E_{PM2.5,a,f,t} \times f_{OC,a,f,t} \tag{3}$$

**2.1 Activity rates and technology division**
Each sector considered in the study (residential, industrial, commercial, agricultural and transport) is classified into coherent sources broadly termed as sub-sectors, listed in Table 1. Activities and their combustion technologies pertaining to each sub-sector are identified for quantifying final energy consumption and resultant emissions. The following section describes in brief the methodological
approach for estimating the fuel consumption in each sub-sector.

*Residential:* For the residential sector, *National Census 2011* provides primary data on the number of households using different types of cooking fuel and sources of lighting for 3915 Village Development Committees (VDCs) and 58 municipalities, i.e., administrative units in rural and urban regions,
respectively (CBS, 2012). The amounts and types of fuel consumed for cooking are based on previous studies reporting the 'useful energy' required for preparing daily food in various utensil-stove-fuel combinations in different seasons (Pokharel, 2004; Khandel et al., 2016). This useful energy is quantified



using the efficiency of the respective cookstove and heat content of the fuel (CES, 2001; Johnson et al., 2013). For lighting, the amount of kerosene used is calculated using the average burn rate, hours of daily usage and number of lamps per household (Lam et al., 2012; DECP, 2014). During hours of power cuts, the fraction of electricity users switching to kerosene lamps was determined by presuming equal weighting assigned to alternate sources such as kerosene lamps, solar lamps, batteries and inverters, and candles and also by constraining the kerosene estimate reported by the Water and Emission Commission Secretariat (WECS) for lighting (WECS, 2014).

For water heating and boiling, the specific energy required to raise the temperature from ambient ($t < 20^{\circ}$ C) to $43^{\circ}$ C was calculated assuming 15 l capita-day$^{-1}$ amount of bathing water (average capacity of commonly used buckets in Nepal). The monthly mean minimum temperature maps are used to identify the districts where the morning temperature falls below $20^{\circ}$ C (MoSTE, 2013). Table 2 provides details on the assumptions for estimating amount of specific energy required to heat water for bathing. Similar temperature threshold ($t < 20^{\circ}$ C) was adopted to estimate the amount of fuel used for space heating that takes place inside and even outside the house. For space heating inside the house, the excess amount of fuel required to warm the space after cooking (i.e. the cooking fire is continued for the purpose of space heating) during winter season was attributed to this activity (Khandel et al., 2016) (Fig. S1). For heating outside, two different groups of population were identified, the first category that works during night shifts hours, i.e., during 11 PM–7 AM and the second category is the resident population who uses space heating after sunset in the evening, i.e., during 7–11 PM. Based on routine observation and expert judgement, number of hours, fraction households engaged and number of days in each month considered are shown in Table 3. Similarly, National Census 2001 is used for activity rates for year 2001. For years in between 2001 and 2011, the data was interpolated using compound annual growth rate (CAGR) which was further extrapolated till 2016 in order to provide trends from 2012 to 2016.

*Industry:* In the industry sector, the information available in the *Census of Manufacturing Establishment* (CME) *2011* report was used for fuel estimation. CME 2011 has surveyed a total of 4,076 industrial units nationwide covering small, medium and large industries (CBS, 2014). The CME questionnaire survey





focused on collating the annual details on proprietorship, organizational structure, production, sales figure, fuel, electricity consumption and details on pollution control equipment. The Department of Industry (DoI), Nepal categorizes these industries into small, medium and large industries based on the economic output (GoN, 1992), whereas this study has categorized industries based on the reported energy use. A total of 1,512 industrial units including cement manufacturing, basic iron, structural metal, brick production, grain mill, noodles, tea and coffee, and pharmaceuticals in this study are considered as large point sources or heavy industries (LPI); whereas rest 2,564 industrial units are considered as small and medium industries (SMI). The fuels in the above LPI and SMI of paper, sugar, beverage, dairy and soap were corrected using specific energy consumption (SEC) from a survey conducted for selected industries and the production data (PACE Nepal, 2012; CBS, 2014). The industrial census is conducted every five years, hence the reports are available so far for 2001/02 and 2006/07. The methodology described above is followed to estimate fuel consumption for these respective years, and interpolated for intermediate years.

*Commercial:* The commercial sector includes all service providing institutions that are largely dispersed over the whole country (Table 1). These institutional units mainly require energy in the form of electricity while few sub-sectors like hotels, restaurants and barrack canteens also consume energy for cooking and other utilities such as water boiling and space heating. The country heavily relies on captive power generator sets using diesel (Gensets) for electricity supply during the main power outages, especially during dry and long non-monsoon season. The amount of diesel use in generators was estimated based on the demand-supply of electricity by the Nepal Electricity Authority (NEA) and the electricity sales in each sector for year 2011 (Fig. S2). For the rest of the years, the diesel consumption was estimated proportionally using respective actual hours of load shedding for each year provided by NEA. With regards to tourism-grade hotels, the total energy use was estimated based on the energy consumed per room for tourism hotels (PACE Nepal, 2012) and their availability during the study period (Tourism Statistics, 2012). A past study on the restaurants in India (FHRAI, 2004) has provided how frequently the urban population goes to dine out in the restaurants; assumed here for the population in urban locations. This frequency along with the energy per capita per day required for cooking is used for estimating the



amount of fuel use in restaurants. For barrack canteens the energy requirement is compiled from the WECS report since no sufficient information is available for their population and activities.

*Agriculture:* The agriculture sector in this study includes activities of agricultural residue burning, use of

diesel pumps, tractors, power tillers, engines for threshers and fugitives of methane from enteric fermentation and manure management. The methodology to estimate the actual amount of waste available for burning after its utilization is followed from Venkataraman et al., (2006). The amount of waste generated after harvesting of crops is estimated using crop production at each district and grain-to-waste ratio for 15 different crops (Table S1) categorized as cereals, pulses, fibers, oilseeds and sugarcane

(MoAD, 2011). This generated waste is further utilized as animal fodder, cooking fuel, industrial fuel and roof thatching and the rest is burned on-field. The amount of animal fodder was estimated using per capita daily requirement compiled from Vekataraman et al., (2006) for different livestock which consists of 40 % crop residue used as fodder in Nepal (Yadav, 2013). Similarly, 2 % of rice straw is assumed to be used for roof thatching (Ravindranath and Hall, 1995; Venkataraman et al., 2006).

The energy required for pumping water was estimated using the amount of water used for irrigation in Nepal, reported by the Food and Agriculture Organization (FAO), and the fraction of people relying on surface water (SW), shallow tubewell (STW) and deep tubewell (DTW) for irrigation (Frenken and Gillet, 2012). The fraction of diesel irrigation pump users relying on surface water, shallow tubewell and deep

tubewell is well established in Nepal, where each consumes different amounts of energy depending on varying pressure head (ADB, 2012). These primary data was used to estimate equivalent amount of energy required to pump the water from varying depth (Fig. S3). For mechanized farming, the *Ministry of Agricultural Department* reported the number of land holdings using tractors, tillers and threshers in 2011 (MoAD, 2011). The total amount of fuel used for land preparation was estimated using area cultivated by

tractors/tillers and diesel consumed per hectare (Baruah and Bora, 2008; Bohra and Kumar, 2015; Erenstein et al., 2008; Kumar et al., 2013) (Fig. S4). In this study, conventional tillage practice is assumed for land preparation that involves four tillage and two planking operations. Similarly, the agriculture statistics reports the use of threshers in each district (MoAD, 2011). The amount of fuel for threshing is



estimated using the diesel consumption rate for threshing achieved through power tiller and small engines (Kumar et al., 2013). The methane emissions from livestock enteric fermentation and manure management are estimated using a Tier-I approach, which involves annual national number of cattle and buffaloes, including dairy and non-dairy categories, along with inclusion of sheep, goats and pigs (MoAD, 2011).

*Transport:* The activity rates for transport sector includes age-distributed vehicle population, fuel efficiency (FE) and vehicle kilometer travelled (VKT) for total eight categories of on-road and off-road vehicles (Shrestha et al., 2013; DoTM, 2013). The actual number of on-road vehicles is modeled using the long-term vehicle registration data (from ca. 1989 till 2016) and the survival fraction of vehicles in each category in every year (DoTM, 2016; Baidya and Borkenkleefeld, 2009; Yan et al., 2011). The survival fraction was modeled using a logistic function to estimate the survival function parameters 'α' and '$L_{50}$' which describes the onset of retirement and the age when 50 % of the vehicles have retired (Yan et al., 2011; Pandey and Venkataraman, 2014). Essentially, the age distribution required for survival parameters was available for two-wheelers and buses for Nepal (for year 2010) from Shrestha et al., (2013) while for the remaining categories they were considered similar to vehicles in India (Pandey and Venkataraman, 2014). The fuel efficiency for each vehicle category was compiled from the survey-based studies conducted in Nepal (Dhital and Shakya, 2014; Dhakal 2003; Bajracharya and Bhattarai, 2013; Pradhan et al., 2006). The VKT was modeled using the survey study in the Kathmandu Valley for two-wheelers, buses, vans and taxis by Shrestha et al., (2013). For the remaining categories these were considered from other literature (Dhital and Shakya, 2014; Dhakal, 2003; Bajracharya and Bhattarai 2013). The emissions from the aviation sector are not considered in this study, as we tend to understand the surface emissions sources within the vicinity of the national boundary.

This study is focused on technology-based emission that included mostly 'contained' burning of fuels. The emission from open burning of municipal solid waste is also prevalent in South Asia. This is an important but yet under-characterized emission source in many developing countries, including Nepal. The emission from open burning of municipal wastes is considered in a companion study by Das et al.,



(2018). Furthermore, we did not consider emissions from forest fires, fugitive emissions from road dust, paddy fields, chemical fertilizer use and biogenic emissions.

## 2.2 Combustion technologies

Understanding the combustion technologies in each sub-sector and characterizing emissions at activity level is one of the principal objectives of this study. Table 4 highlights the technologies considered in each sector and sub-sector.

In residential cooking, the type of cookstove used is mostly governed by the type of fuel burned. A large

fraction of rural households in Nepal still relies on traditional mud cookstoves (TCS) that use firewood, animal dung cake and agricultural residue as fuel. Apart from traditional cookstoves, 1.26 million improved cookstoves (ICS) are disseminated and 0.3 million biogas plants were installed until 2011 under various renewable energy technology (RET) programmes by the Alternative Energy and Promotion Centre (AEPC) of the government and the Centre for Rural Technology, Nepal (CRT/N) (AEPC, 2012).

For water heating and boiling, population in urban areas uses LPG stoves and kerosene stoves, whereas in rural areas they rely on firewood in traditional cookstoves (WECS, 2014; CBS, 2012).

In the industry sector, energy is required for thermal purposes and utilities. Industries like tea, coffee, pharmaceuticals, noodles and other small scale industries consume solid fuel such as coal, wood and rice husk, only to be used in furnaces and boilers for thermal energy and for steam generation. The iron and

20 steel industry in Nepal converts the imported billets into elongated rods using rolling mills that heavily consume furnace oil (FO) in reheating furnace. The cement industries in Nepal are mostly grinding units with 14 mine based units equipped with rotary kilns (Pandey and Banskota, 2008). The grinding units feed on electricity either from the grid supplied by NEA or generated in-house using diesel generators. The mine-based units consume coal and other raw material in horizontal/vertical rotary kilns. The

25 combustion technologies in industries like sugar, paper, beverages and dairy farms are identified based on an extensive survey that reports the process and the energy consumption patterns for those industries (PACE Nepal, 2012). The brick kilns are mostly fixed chimney Bull's trench kilns (FCBTK) either with straight firing or zig-zag firing technology and a small fraction of clamp kilns (CK) and the vertical shaft



brick kilns (VSBK). A total of 609 brick kilns, whose geolocations were identified, are considered in this study out of which 557 are FCBTK and 52 clamp kilns during 2011.

In the commercial sector, since the activity involves energy use for electricity purposes, diesel generators are considered to be the prominent combustion technology in this sector. In the case of restaurants, a technology division similar to residential sectors is followed, with additional boilers required in tourism hotels for hot water generation. The combustion technologies in the agriculture sector include diesel use in irrigation pumps, tractors, power tillers and threshers. Since insufficient information is available in terms of size for diesel pumps, tractors, power tillers and threshers, the fuel consumption estimates do not account for any additional factors that would have led to more accurate (new) fuel estimates in this study.

In the transport sector, a total of eight vehicle categories are considered including an off-road category for tractors and power tillers (Table 4). The tractors and tillers are used with trailers for transportation purposes during non-farming days. Since more than 80 % of the vehicles are imported from India, we have assumed all the vehicles comply with Bharat Standard (BS) norms for emissions estimation. Diesel vehicles like jeeps/taxis, mini-bus, microbus and bus are treated as public passenger vehicles, while mini trucks, pick-ups and trucks are treated as public freight vehicles. Around 40 % of the diesel vehicles in Nepal are categorized as super-emitters or high emitters due to poor maintenance of vehicles, old vintages, add a large fraction of shoddy roads (Bond et al., 2004; Yan et al., 2011; personnel communication).

## 2.3  Emission factors

In residential cooking, the emission factors (EFs) for biogas stoves, traditional cookstoves (TCS) and improved cookstoves (ICS) using firewood and dung cakes are mostly taken from recent field measurements during the NAMaSTE campaign (Nepal Ambient Monitoring and Source Testing Experiment) in Nepal (Stockwell et al., 2016; Jayarathane et al., 2017) except for the ICS $PM_{2.5}$ and OC which are taken from Jaiprakash et al., (2016) and biogas stove $PM_{2.5}$ from Smith et al., (2000) (Table S2). The OC EFs for the TCS using firewood and dung cakes were averaged with other studies since the inorganic fraction imbalances the total $PM_{2.5}$ emissions. In the NAMaSTE campaign, $CO_2$, $CH_4$, NOx,



CO, NMVOC, $PM_{2.5}$, BC and OC EFs were measured explicitly from TCS and ICS reflecting the regional practice and emission factors. For kerosene and LPG stoves, the EFs are compiled from lab-simulated measurements by Smith et al., (2000) for $CO_2$, $CH_4$, $N_2O$, NOx, CO and NMVOC, and averaged from Habib et al., (2008) and Smith et al., (2000) for $PM_{2.5}$ whereas the fractions of BC and OC were taken

from Habib et al., (2008). The GHGs and NMVOC EFs for agricultural residue burned in traditional cookstoves were compiled from Smith et al., (2000); for aerosols and CO, they were taken from in-field measured Indian EFs reported by Pandey et al., (2017), and the EF for NOx was assumed similar to the saw dust from NAMaSTE campaign (Stockwell et al., 2016). $SO_2$ EFs for burning TCS and ICS were considered from Habib et al., (2004), for the LPG and biogas stoves based on the sulphur content; for

kerosene stoves from Zhang et al., (2000), who reported values for the Chinese wick and pressure stove. For kerosene lamps, the measured EFs from Lam et al., (2012) were available for all pollutants except $CH_4$ and NMVOC, which were considered to be similar to kerosene stoves. For biogas lamps, all EFs were considered similar to the biogas stove. In regards to space heating, the EFs for open burning of firewood were assumed similar to three-pot cookstoves, whereas the EFs for dung cakes were measured

during NAMaSTE campaign (Stockwell et al., 2016; Jayarathane et al., 2017).

For the industry sector, technology-linked EFs were used from the EPA AP42 repository that has identified the combustion and process activities for different sources and industries (Table S3). The EFs for brick production were compiled from Weyant et al., (2014) for FCBTK in South Asia. Stockwell et

al., (2016) measured EFs for the zig-zag firing in FCBTK and clamp kilns in Nepal. These EFs of $CO_2$, CO, BC and OC from NAMaSTE campaign were averaged with Weyant et al., (2014) for similar kiln types in order to reduce the uncertainties in EFs. Whereas the EFs for $CH_4$, $NO_X$ and NMVOC were taken from Stockwell et al., (2016). $SO_2$ EF for FCBTK (straight firing) was assumed to be similar to coal stoker and for the zig-zag firing and clamps kilns from Stockwell et al., (2016) (Table S4).

Commercial sector activities closely resemble those in residential cooking and hence the EFs are similar. For diesel generators, the recently measured EFs from the NAMaSTE campaign were considered for all pollutants except CO and $SO_2$ (Table S5). The EF for CO was averaged from data provided in Shah et



al., (2006), since of the two diesel generators measured during the NAMaSTE campaign, one apparently reflected steady-state conditions and was regularly maintained, while the other one appeared to be improperly regulated as it provided extremely high values. The $SO_2$ is estimated using the sulphur content of the fuel with no retention.

Similarly, for irrigation pumps, the EFs for all pollutants were compiled from the NAMaSTE campaign, except for $N_2O$ with $SO_2$ (Table S5). For mechanized tractors, power tillers and threshers, the EFs were compiled from a study that reports the EFs for off-road vehicles measured across different power capacity (Notter and Schmied, 2015). Region-specific EFs were considered for rice straw and cereal residue open

burning from the NAMaSTE campaign (Stockwell et al., 2016) (Table S6). For pulses, oilseeds, jute crop and sugarcane reside, the NAMASTE campaign crop residue emission factors were averaged with on-field measured emissions factors from Turn et al., (1997), Kim Oanh et al., (2011), Li et al., (2007) and Andreae and Merlet, (2001). The methane emission factor from enteric fermentation and manure management was considered from Jha and Singh, (2011), who reported the values based on the gross

energy from feed intake and the methane conversion rate for dairy and non-dairy cattle in India.

In the transport sector the EFs for two-wheelers were compiled from the NAMaSTE campaign for $PM_{2.5}$, BC, OC, CO, $NO_X$, $CH_4$ and $CO_2$ (Stockwell et al., 2016; Jayarathane et al., 2017) (Table S7). For other categories, emission factors of $N_2O$, $NO_X$, CO, NMVOC, $PM_{2.5}$ were considered from Shrestha et al.,

(2013) who studied the emissions from the transport sector in Kathmandu Valley using vehicle survey data in the International Vehicle Emissions (IVE) model. However the NOx and $PM_{2.5}$ emission factors for heavy diesel vehicles like trucks and buses from Shrestha et al., (2013), were way too high compared to other literature (Zhang et al., 2009; Sadavarte and Venkataraman 2014; Streets et al., 2006). Therefore, they were considered from Sadavarte and Venkataraman, (2014) which are the values for Indian vehicles

modelled using the MOBILEv6.2 model, also consistent with values from Zhang et al., (2009) for Chinese vehicles. The NMVOC emissions from Shrestha et al., 2013 also include running evaporatives, and reflect real world emissions under increasing ambient temperature. The $CO_2$ emission factors for all categories of vehicles, other than two-wheelers, were considered from a chassis dynamometer study that measured



values for different vintages using an Indian driving cycle (ARAI, 2007). Fractions of BC and OC for all categories of vehicles were taken from Bond et al., (2004). $SO_2$ emissions were calculated using the sulphur content of BS-II/III/IV fuel imported from India with no retention.

## 3. Result and discussions

### 3.1 National energy estimates and sectoral fuel consumption

Using the methodology described above in each sector/sub-sector and the efficiency associated with fuel-technology combination, the total national energy consumption is estimated explicitly for the base year 2011 and also for each year over the period 2001-2016.

*3.1.1 National energy trend*

Figure 2a and 2b show the proportional contributions of each sector and fuel type, respectively, to the total energy consumption in Nepal in 2011. It should be noted here that as our study focuses on technology-based emissions, we did not include the electricity (supplied by the hydropower stations) consumption in the total energy consumption. Previous studies report that the contribution of electricity to total energy consumption in Nepal is less than 5 % (WECS, 2010; WECS 2014). We estimate that Nepal consumed 378 PJ of energy in 2011, and it can be observed that the residential sector is the highest and dominant consumer of total national energy consumption (79 %), followed by industry (11 %), transport (7 %), commercial (2 %) and agriculture (1 %). Figure 2c shows the trend of the energy consumption at sectoral level for each year from 2001 to 2016, with national total energy consumption of 463 PJ in 2016, a factor 1.62 increase compared to 286 PJ in 2001. The majority of the energy is derived from solid biomass (including firewood, agricultural residue and dungcakes) that accounted for 91 % of the total energy consumption in 2001, which has decreased to 84 % in 2016 (Fig. S5). The imported fossil fuel contributed 9 % to the national total energy consumption in 2001 which increased to 16 % in 2016.

The energy consumed in the residential sector was estimated at 322 PJ in 2016, a factor of 1.32 higher than in 2001. In this sector activities of cooking contribute to 68 %, space heating 20 %, water heating 12 % and a small amount from kerosene lighting to the total residential energy use throughout the analysis



period 2001–2016. The national population has almost doubled (a factor of 1.8) in the same period. There was continuous increase in the number of households (1.27 million households) using energy efficient improved cookstoves (ICS) till 2016 (AEPC, 2012). Also, the amount of kerosene usage in 2016 for lighting has reduced by 50 % compared with 2001 due to an increase in the number of households with

access to electricity (5.68 million in 2016) and greater penetration of renewables like solar household lighting systems (0.6 million units till 2016) (AEPC, 2012). These more energy efficient interventions helped offset the emissions they would have otherwise been.

The industry sector consumed 79 PJ of energy in 2016, a 3 times increase since 2001. The point source

industries consumed 77 % of the industrial energy use in 2001 that further increased to 84 % in 2016. Basically, the number of manufacturing establishments in area source industries has increased from 2163 to 2564, while that in point sources industries has increased from 1050 to 1512 during the period 2001– 2016 (CBS, 2014).

The commercial sector consumed 6.4 PJ of energy in 2016, which is almost double (a factor of 1.8) the energy used in this sector in 2001. Along with an increase in population and number of tourists, the use of diesel in captive power generators to cope with shortage of power supply that started becoming serious since 2006 is the foremost source that consumed one-third of the commercial energy in 2016. During 2001–2005, the load shedding in Nepal was insignificant and ranged from 1 to 8 GWh. However, this

problem increased gradually from 2006 reaching to a short fall of 1000 GWh in 2010. The load shedding worsened further from 2011 to reach a shortage of 1300 GWh in 2015, with a sharp decline to 474 GWh (modelled using electricity demand, actual sales and electricity that can be supplied) in 2016 as a result of careful power management measures implemented by the Nepal Electricity Authority.

The transport sector, comprising of both on-road vehicles and off-road tractors and tillers, consumed 45 PJ (with additional 6 PJ in aviation) of energy in 2016, a factor of 4.5 higher than in 2001. This increase in fuel can be attributed to continuous increase in total number of fleets from 0.27 million to 2.23 million





during the study period (DoTM, 2016), which is estimated using the vehicle registration data and the survival function for each category of vehicle.

The amount of agricultural residue burned on-field was estimated at 5.8 million tons in 2001, which has increased to 8.0 million tons in 2016. This increase can be attributed to an increase in annual production of cereals by a factor of 1.19, pulses by 1.47, oilseeds by 1.67 and sugarcane by 1.86 between 2001 and 2016. Out of the total waste generated, 35 % was used as animal fodder, 3 % industrial fuel, 17 % cooking fuel, 2 % thatching and 43 % waste unutilized was burned on-field.

If the trend in national fuel consumption is analyzed with respect to 2001, it can be inferred that the import of fossil fuels has increased many-fold more than the increase in solid biofuels in the country (Fig. S6, S7; Table S8). The LPG consumption in Nepal has increased by a factor of 6.5 over its 2001 use, mainly because a large population shifted to a cleaner fuel for cooking in this period. Even the consumption of gasoline has steadily increased 6 times due to increases in the import of gasoline vehicles; similarly diesel increased by a factor 4, as diesel was also heavily used in diesel power generators, besides diesel vehicles. It is interesting to observe here that there was a jump in diesel consumption in the year 2009 when there was a policy shift equating the kerosene price with diesel. This resulted in a sharp fall in kerosene consumption in 2009 and continued to fall gradually afterwards (Fig. S7). Before this policy intervention, kerosene was used alternatively for diesel (in fuel adulteration). A low growth (a factor 1.1–2.1) is observed in solid biofuel use mainly due to higher penetration of improved cookstoves and cleaner fuel such as LPG for cooking and increase in renewables (e.g., biogas).

*3.1.2 Comparison with WECS' 2011 estimation*

The national fuel and energy consumption in individual sectors estimated in this study for the year 2011 are compared with the national totals reported by the Water and Energy Commission Secretariat (WECS, 2014) as shown in Fig. 3. According to WECS, a total 376 PJ of energy was consumed in 2011 in all forms such as hydro-electricity, thermal-electricity, other renewables, fossil fuels and solid biofuels. However, a one-to-one correspondence for each activity and fuel is made while comparing the energy, which shows 378



PJ of fuel energy estimated in this study against 329 PJ (energy consumed in combustion activities) by WECS. In present study a small increase in energy consumption in the residential sector (300 PJ) can be observed compared to the WECS' estimate 265 PJ, which can be attributed to the inclusion of agricultural residue as a fuel for cooking not considered in the WECS' estimate.

In the industry sector, the present study estimates 41 PJ of energy consumption which is 59 % higher than that reported in the WECS' report (26 PJ). This increase is attributed to biomass in the form of rice husk taken into account in this study as boiler fuel in paper, sugar, alcohol, soap and noodles industries (PACE Nepal, 2012) using specific energy consumption (SEC). Similarly, consumption of coal in brick production and cement manufacturing was also corrected using SEC. Apart from these fuels, the furnace oil (FO) use in the metal industry and captive power generation, which was completely missing in the WECS' detailed fuel and energy consumption data, was considered in this study. Basically, Nepal Oil Corporation (NOC) is an authorized agency to import fossil fuels (all of kerosene, gasoline, diesel and liquefied petroleum gas (LPG), and FO) and made available through public distribution system. However, in recent years, FO was also imported as an industrial fuel by private importers, without having a record by the NOC, and hence not reported under import statistics by the NOC. Therefore, the FO reported in this study accounts for the total import available from foreign trade statistics report which was not accounted for by NOC (MoF, 2012) and WECS' report.

The energy consumption in commercial hotels and restaurants was estimated using energy consumed per room and energy per capita as described in Sect. 2.1. The energy estimated was a factor of 0.73 compared to the WECS. This difference can be explained by higher use of firewood (factor 3.6), LPG (factor 2.4) and kerosene (factor 2.6) in WECS' report. The commercial sector also relies on diesel generator sets during load-shedding hours, which seems to be completely missing in the WECS' estimate. 4.6 PJ of diesel was estimated to have been used in generator sets, that accounts for 58 % of the energy consumed in the commercial sector in 2011.





In the agriculture sector, the amounts of diesel consumed for irrigation water pumps, tractors, power tillers and threshers are in agreement with WECS' estimates. In the transport sector, the energy estimated in this study was 5 % lower than the WECS's estimate, especially in diesel vehicles. For gasoline vehicles, the alpha ($\alpha$) and $L_{50}$ were modelled using the registered population for two-wheeler motor-cycles and age distribution from Shrestha et al., (2013), while the remaining gasoline was distributed among four-wheeler cars (due to non-availability of the age distribution) closing the annual consumption, thereby similar to amount of gasoline reported in the WECS' report. The on-road/off-road diesel vehicle fleet was modelled using a logistic function, chosen similar to the case of vehicles in India (Pandey and Venkataraman, 2014) except for the buses for which the age distribution was available from Shrestha et al., (2013).

### 3.1.3 Kathmandu Valley and Nepal

The Kathmandu Valley is a highly urbanized area and more advanced than the rest of the country and varies significantly in energy-use patterns and their respective emissions. The following section compares the energy consumption in the Kathmandu Valley (KTM) and all of Nepal (NPL) during 2011 across different sectors and fuel types. Figure 4a and 4b show the estimated total energy consumption in KTM as 30 PJ, which is 8 % of the national energy consumption (378 PJ). In the Kathmandu Valley, the residential sector shares 41 % of the total energy consumption, followed by transport (26 %), industry (22 %) and commercial (10 %), with a small amount in agriculture sector which in comparison do not resemble the national energy consumption pattern across these sectors.

The intensity of activity rates in each sector for the Kathmandu Valley is different from the whole country. If the sectors are disaggregated further to understand the fuels contribution, it is found that the residential sector in the Kathmandu valley consumes 73 % solid biomass, 27 % LPG and small fraction of kerosene while all of Nepal consumes 98 % solid biomass and 2 % LPG. The Kathmandu Valley is home to only 10 % of the national population where 84 % of the households use LPG for cooking, versus only 21 % nationwide that leads to 41 % of the national LPG consumption in the Kathmandu Valley alone. Also in the case of sources of lighting in the Kathmandu Valley, 98 % of households have electricity as the



primary source with only 1 % kerosene users, which is different from only 67 % households using electricity and 18 % kerosene in Nepal.

Nearly one-fourth of manufacturing establishments (947 out of 4067 industries) in Nepal are located in the Valley, which includes 109 point source industries (including 76 brick manufacturing units reported in CME 2011 report) and 838 area sources. From the commercial sector, 503 tourist hotels, 363 academic campus, 394 financial institutions and 78 hospitals are in the Kathmandu Valley, out of 817 tourist hotels, 1101 academic campus, 1245 financial institutions and 143 hospitals in Nepal. The use of diesel and LPG especially in tourist hotels and diesel generator sets in commercial sector in the Kathmandu Valley makes a huge difference in fuel fraction. Approximately one-third of national energy in the transport sector is consumed by vehicles registered in the Bagmati Zone alone. Around 30 % of the national vehicles are registered in the Bagmati zone that includes three districts in the Kathmandu Valley. This leads to 50 % of national gasoline and 27 % of national diesel consumption in the Valley.

In 2016 the Kathmandu Valley consumed 51 PJ of energy which is a factor of 1.7 higher than 2011, with an increase in energy from mainly industry and transport sector (Fig. S8). In spite of an increase in population, the residential sector has seen growth in use of energy efficient renewable technologies and cleaner fuels (e.g., LPG) which didn't cause a spike in 2016 energy compared to 2011. Similarly, the increase in production by co-located point source industries and vehicle registration have led to ramp up the sectoral energy in these two sectors by a factor 3 and 2, respectively. The energy (fuel-based) in the commercial sector has reduced by 50 % due to a reduction in load shedding, ultimately decreasing the diesel consumption by small but numerous captive diesel power generators.

### 3.2 Emission trends and sectoral contribution to national estimates

The emissions were estimated for 10 pollutants, $CO_2$, $CH_4$ and $N_2O$ (*GHGs*), $NO_X$, CO, NMVOC and $SO_2$ (*trace gases*), and $PM_{2.5}$, BC and OC (*aerosols*) using the technology-linked EFs in each sub-sector. The following section explain the emission estimates for 2011 in each sector and sub-sectors (Fig. 5a) and the emissions trend (Fig. 5b) in order of aerosols and its precursor, trace gases and greenhouse gases.



For analysis and comparison purposes, only the combustion based emissions from energy sources are considered, leaving out fugitive emissions from livestock management.

In 2011, 239 Gg $PM_{2.5}$, 28 Gg BC and 99 Gg OC emissions were estimated from five energy-use sectors. Similar to the total energy consumption pattern, the residential sector contributes the highest share to particulate emissions (78 % $PM_{2.5}$, 76 % BC and 80 % OC). Within the residential sector, the activities of cooking and space heating are responsible for ~70 % emissions of $PM_{2.5}$, BC and OC. Additionally BC emissions are also emitted from residential kerosene lighting (4 %) using cotton wick kerosene lamps and kerosene lanterns. Industrial sources contribute to 2.3 % $PM_{2.5}$, 8.2 % BC and 1 % OC with significant emissions from brick kilns (FCBTK) and use of firewood and ricehusk in industrial furnace of sugar, beverages, noodles and other small industries. Nationally, the transport sector contributes to only 1.1 % $PM_{2.5}$, 5.1 % BC and 0.8 % OC emissions largely from diesel freight and public passenger vehicles. A fleet fraction of 0.40 was assumed to be super-emitters (a higher fraction than used in a study of transport sector emissions in India, due to vast complex terrain and no proper policy to phase out older vehicles in Nepal) which under poor maintenance emit emissions as high as 10 times that of a normal vehicle (Pandey and Venkataraman, 2014). The emissions from the agriculture sector contributed to 18.3 % $PM_{2.5}$, 10.3 % BC and 17.2 % OC, with more than 90 % from open burning of agricultural residue.

Emissions of 28 Gg $SO_2$, 72 Gg $NO_X$, 1984 Gg CO and 477 Gg NMVOC were estimated nationally in 2011. The $SO_2$ emissions were estimated using the sulphur content of the fuel with no retention in liquid and gaseous fuels. The agriculture sector, especially residue burning contributes 47 % of $SO_2$ emissions (from residue burning), followed by 40.7 % from industry sources, especially from the coal users such as brick kilns and cement manufacturers. The residential sector is the third largest gross emitter of $SO_2$ adding 11.8 % to the national total, and 0.5 % each from the commercial and transport sectors. Again, the residential sector overshadowed all others and contributed significantly to 46.8 % $NO_X$, 78.2 % CO and 80.7 % NMVOC. Residential cooking, space heating and water heating are the major sources of incomplete combustion of biomass fuels and hence high CO and NMVOC emissions. After residential emissions, the transport sector contributes 20.2 % of $NO_X$. High NOx emissions were emitted by diesel





passenger (8 %) and freight vehicles (9.4 %). The industry sector was responsible for 15 % $NO_X$, 2 % CO and 0.5 % NMVOC emissions. The clinker production in cement industries has a large share of $NO_X$ emissions, which was about 60 % of the total point source industries. The agriculture sector emitted 14 % $NO_X$ with the second largest contribution to CO (13.7 %) and NMVOC (14.6 %) after residential, since

9.3 million tons of residue was burned on-field in 2011. Nationally, 8.4 Tg $CO_2$, 666 Gg $CH_4$ (including 547 Gg from livestock fugitive emissions and 119 Gg from energy sources) and 2.5 Gg $N_2O$ were estimated in this work. A non-renewability factor of 10 % for firewood and 0 % for agricultural residue and dung cakes was considered while estimating $CO_2$ emissions following Venkataraman et al. (2010), reducing the $CO_2$ contribution from the residential sector, while still responsible for 89 % of $CH_4$ (only

combustion based anthropogenic sources are considered in the $CH_4$ analysis) and 77 % $N_2O$ based on the energy consumption. The industry sector emitted 46 % of national $CO_2$ emissions mostly driven by energy from fossil fuel and biomass in brick and cement manufacturing, and 14.3 % from transport sector.

Figure 5b shows the normalized trend in emissions with respect to 2001. Overall there is an increase in

emissions with a factor ranging from 1.2 for species like $CH_4$, $PM_{2.5}$ and OC to a maximum increase by a factor 2.4 for $CO_2$ and 2.1 for $NO_X$ between 2011 and 2016. The change in slope and increase in $CO_2$ and $NO_X$ emissions can be well explained by a sharp increase in import of petroleum fuels like LPG, petrol and diesel after 2008. The emissions of $PM_{2.5}$, OC, NMVOC, CO and $SO_2$ have increased by a factor 1.2–1.4 following some ups and downs. The following changes in $SO_2$ emissions can be explained

by the decreasing sulfur content in the import of petrol and diesel, that is, a decrease from 500 mg sulfur per kg fuel (BS-II norm till 2005) to 50 mg sulfur per kg fuel (BS-IV norm 2011 onwards). The increasing trends in $PM_{2.5}$, OC, NMVOC and CO follow the energy use in the residential sector, while the changing sinusoidal pattern (ups and downs) reflects the increase and decrease in agricultural crop production.

**3.3    Technology based emissions estimates**

The emissions are characterized for 36 different combustion technologies using their respective emission factors. The following section describes the top six combustion technologies and their overall contribution to the 2011 national emissions estimates of aerosols, ozone precursors and greenhouse gases.





For aerosols, the top six combustion technologies correspond to 95 % $PM_{2.5}$, 88 % BC, 96 % OC and 92 % $SO_2$. The high $PM_{2.5}$ and OC emissions are estimated from use of firewood, dungcake and agricultural residue in traditional cookstoves; open burning of wood and dungcakes for space heating activities and agricultural residue burning (Fig. 6). Various national and international organizations have promoted the use of improved cook stoves (ICS) that implicates energy efficient stoves with improved air quality and health co-benefits. So far, 1.26 million ICS disseminated during 2000–2011 are considered in this study which has reduced 13 % $PM_{2.5}$, 12 % BC and 13 % OC emissions nationally compared to the emission scenario without ICS distribution. In the case of BC, apart from above combustion technologies, high amounts of BC are also emitted from FCBTK in brick production and kerosene lamps (Fig. 6). The coal combusting technologies like FCBTKs and cement kilns emit about 35 % of $SO_2$, while the main fraction of 47 % is emitted from open burning of crop waste, 5 % from traditional stoves using wood and agricultural residue that make 92 % from the top six polluting technologies.

In the case of ozone precursors, apart from residential and industry sectors, we start to observe the listing of polluting technologies from transport sector too. Combustion technologies like traditional cookstoves using firewood and agricultural residue emit 32 % NOx, while heavy duty diesel passenger (buses) and freight (trucks) vehicles emits 16 % and agricultural residue 12 % and cement kilns 8 % that accounts for 68 % of the national estimate (Fig. 6). The top polluting technologies are the TCS from residential sector (cooking and water heating) in all pollutants, since the residential sector is largely driven by energy consumption compared to the diesel use in trucks and buses. CO, a tracer of incomplete combustion is largely emitted from biomass burning with TCS that adds up to 63 %, while 26 % is emitted from opening burning of wood and crop residue. Gasoline vehicles also contribute to 3.3 % of CO which aggregates to a total of 93 % of the national estimate. Recent years have seen a sharp increase in use of two-wheeler vehicles, with about 1.03 million two-wheelers registered till 2011 (2.18 million by 2016) nationwide, which are estimated to consume 113 million litres of gasoline. NMVOCs emissions are completely driven by combustion of biomass in every form of fuel in TCS, and also in ICS, which contributes 94 % of the



total NMVOCs. Replacing 1.26 million TCS with ICS has reduced NMVOC emissions by 15 % nationally.

High emissions of $CO_2$ (considered after non-renewability factor) are largely driven by high energy consumption in cement kiln (28 %), TCS using wood (16 %), brick kilns (12 %), LPG stove users (7 %), diesel trucks (6 %) and open burning of wood (6 %) that makes 75 % of the national estimate. Only pyrogenic emissions are considered for analysis and hence the $CH_4$ and $N_2O$ emissions follow the top polluting technological trends, similar to $PM_{2.5}$, CO, NMVOC and OC, i.e., TCSs and open burning of firewood, agricultural residue and dungcakes.

## 3.4 Emissions comparison

### 3.4.1 MIX and REAS emission inventory

The present study NEEMI emissions are compared with the Asian anthropogenic MIX emission inventory which was prepared for the MICS-Asia and the HTAP projects (Li et al., 2015). One-to-one correspondence for sectors and activities are considered while making the comparison. The fugitive component of $CH_4$ emissions from REAS (MIX) is not considered here as it comprises of sources of enteric fermentation, manure management, paddy fields, coal mining and waste which are beyond the scope of the study. Table 5 shows the national emissions ratios of NEEMI to MIX in 2010. It is observed that the NEEMI estimates lower emissions of $NO_X$ (0.73), CO (0.80), BC (0.91), OC (0.77) and $SO_2$ (0.47) while $CH_4$ (1.20), $N_2O$ (1.35), NMVOC (1.06) and $PM_{2.5}$ (1.38) emissions are higher than the MIX estimates. This difference in emissions can be explained by the underlying assumption of fuel allocation and the EFs in each sector. Since the MIX inventory for Nepal was reproduced using the REAS, the following analyses have compared the REAS 2008 sectoral emissions with NEEMI 2008 emissions to explain the difference in emissions (Table S9). In both the inventories, the residential sector dominated all the emissions, emitting more than 90 % of the national estimate which identifies it as the single most influencing sector to have caused low ratios for $NO_X$, CO, BC, OC and $SO_2$ nationally. Further the EFs for biomass burning in residential sector in MIX and NEEMI were investigated and it was found that, the ratio of NEEMI to MIX weighted average EF was a factor 0.89 for $NO_X,$ 0.88 for CO and 0.22 for $SO_2$.



However, for BC the weighted average EF in NEEMI was 1.1, a factor higher than MIX and for OC the EFs were similar. Moreover, the REAS inventory was also compared nationally from 2001–2007, delineating similar ratios as found in 2008 (Fig. 7). This shows the use of invariant emission factors and other underlying assumptions for different time scale in REAS with an increase in fuel consumption from 2001 onwards, while the resultant $SO_2$ emission factor in NEEMI reflects the timely improved sulfur content of diesel and petrol. Apart from the difference in emission factors in biomass fuel, the contribution from the industry sector is weakly estimated for NOx, BC and $SO_2$ in REAS, which is 4.5 %, 1.3 % and 20.6 % compared to 14.9 %, 7.8 % and 73.3 % in NEEMI. This concludes that the difference in emissions is not predominated by EFs only, but that there is also an influence of fuel consumption and its appropriate allocation to combustion technologies. Since these factors are carefully considered in this study, the NEEMI estimates suffer less uncertainty in emissions.

### 3.4.2 Kathmandu Valley and Nepal

The emissions from the Kathmandu Valley (KTM) are compared with all of Nepal (NPL) to understand the relative contributions from different sources in the Kathmandu Valley for 2011. The land use pattern in the Kathmandu Valley is mostly non-agricultural; therefore, the analysis does not include the emissions from agricultural residue burning. There are three administrative districts in the Kathmandu Valley, namely Kathmandu, where the capital city (Kathmandu Metropolitan City, or KMC) is located, Lalitpur and Bhaktapur. Being the commercial and financial hub of the country, the population residing in the valley is about 2.5 Million, ~10 % of the total population of Nepal in 2011 (CBS, 2012). Table 6 shows the emission ratio of KTM to NPL for all pollutants and sectors, which ranges from 0.04 to 0.13, with a minimum for $CH_4$, $PM_{2.5}$ and OC emissions and a maximum for NOx emissions, which reflects strong sources of $NO_X$ in the Kathmandu Valley, definitively high diesel use. These ratios were also estimated for each sector to understand its relative contribution.

On a sectoral level, the emissions (all species) from the transport sector in the Kathmandu Valley are 30 % to 55 % of the national estimates. One of the important reasons for high emissions from transport is the number of vehicles registered in the Bagmati Zone (a zone of eight districts including three districts



in the Kathmandu Valley). 30 % of national fleet is registered in Bagmati Zone, which subsequently results in a high level of fuel consumption. Next, the commercial sector is of importance, for which the emission ratio ranges from 0.10–0.35 for individual pollutants. This sector comprises of the sub-sectors of hotels, restaurants and heavy use of diesel in generators during load shedding hours (power cuts).

According to the Nepal tourism statistics, there are 423 hotels registered in the Kathmandu Valley, compared to 291 outside the Valley. Also the urban population that dines in restaurant is assumed to be a factor 3 times higher than the rural population, which increases the fuel consumption and its emissions. The industry sector emissions ratio for all species ranges from 0.07–0.17, except for $N_2O$, which is 0.26. There are 110 large point source industries within the Valley (out of 1512 nationally) which especially

includes 76 brick production units (identified using geolocations), which tend to use biomass and coal, increasing the $PM_{2.5}$ and OC emissions. Industries that manufacture structural metal also contribute. The residential sector emissions from the Kathmandu Valley are only 3–4 % of the national emissions for all pollutants, in spite of the fact that ~10 % of the population resides in the Valley. Due to extensive urbanization, the population relies on cleaner sources of fuel, viz. 84 % of households use LPG and 10 %

of households use electricity as their primary source of fuel for cooking (CBS, 2012). In addition, < 5 % of the households use sources other than electricity for lighting. In contrast to the valley, 82 % of households rely heavily on biomass (firewood, agricultural residue and dungcakes) for cooking and nearly 35 % of households use sources other than electricity for lighting.

For 2016, the emissions from the Kathmandu Valley were compared with 2011 in order to understand the impact of shifts in technologies on emissions. The total emissions from KTM during 2016 have increased by a factor varying from 1.05 for OC to 2.2 for $SO_2$. Table S10 details the emissions in 2011 and 2016 over the Kathmandu Valley. The two polluting sources (i) straight firing brick kilns and (ii) diesel generator sets were changed to the zig-zag firing, due to rebuilding the brick kilns after the 2015

earthquake, and almost negligible use of diesel in generator sets, due to improvements in load shedding hours during 2016. The shift to the zig-zag firing has resulted in reductions of 65 % CO, 44 % $PM_{2.5}$, 93 % BC and 17 % $SO_2$ emissions from brick production, while the near-complete phasing out of diesel generator sets has reduced all pollutants from diesel generator sets to effectively zero.



### 3.5 Spatial and temporal distribution

High resolution emission estimates require suitable surrogates to distribute emissions spatially according to their activity locations. Each sector and sub-sector activities were tracked down and distributed as point, area and line sources. Figure S9 shows the proxies considered in each sector. The emissions from residential activities were distributed using population surrogates from the Central Bureau of Statistics (CBS) for 2011. This vector data of population available at each Village Development Committee (VDC), an administrative unit in rural areas, and municipalities in urban areas, is processed using dasymetric mapping techniques (in addition to census data use ancillary information like topographic maps and LULC data to generate population density maps) and converted into a uniform raster grid of 1 km × 1 km resolution (Mennis, 2009). Similarly, these spatial proxies were also used in space heating and water heating sub-sectors. Industrial emissions from point sources (Table 1) are distributed to approximate locations identified based on CME survey information (CBS, 2014). Similarly, for emissions from brick production, the exact geospatial locations of the individual ~470 kilns were identified using web URL (Google Earth) and CME reports (Fig. S9).

Industries categorized as area sources were aggregated and their emissions distributed in respective VDCs and municipalities using population surrogates. Emissions from commercial sources of hospitals, hotels, banks and academic institutions were distributed based on their locations at VDCs and municipalities. Emissions from sub-sectors of restaurants, barrack canteens and other service sectors were treated similarly to the residential sector. The census of agriculture reports the irrigation and cultivation activities using tractors, pumps, power-tillers and threshers at the district level. These identified districts, along with land-use-land-cover (LULC) maps of agricultural area, were utilized for distributing emissions from the agriculture sector. The emissions from mobile sources were distributed using proxies of population and road density maps. With a basic understanding of population and its urbanization, the emission from private cars (gasoline) were distributed using urban population densities, while for two-wheeler vehicles, these were spread across rural and urban population maps. For diesel, especially public passenger and freight vehicles, due to insufficient information, a logic of 50–50 % was used to distribute emissions on





population and road density network. Figure 8 shows the final spatial distribution of PM$_{2.5}$, BC, NOx and CO emissions for 2011. As the residential sector dominates the whole inventory, the gradient across different parts of the country mostly reflects the population density, with a few hotspots for point sources.

Figure 9 shows the seasonality of various activities on a monthly resolution basis. Temporal variability in emissions was introduced in the activities of brick kiln production, where the official firing of kilns starts from December and continues for the next 4 months until April, before the onset of the monsoon. Around 87 % of households throughout the country hire tractors and power tillers for tillage operations, which usually accounts for 40–45 days, including 2–3 days of usage on their own farm land. Therefore,

two months in the pre-monsoon season and post-monsoon season are considered for distributing emissions from tractors and power tillers used on-field. The seasonality in agricultural residue burning is accounted for using the number of fire counts from the MODIS instrument onboard the Terra satellite for the respective study period. Emissions from space heating and water heating are considered as per the temperature profiles across each district. Diesel generators were heavily used in the non-monsoon season

during hours of load shedding (before the shift in power management). Therefore, the seasonal variability in the use of diesel generators is followed as per the trend in monthly hours of load shedding reported by NEA. The analysis of the monthly distribution of emissions showed that April constituted the maximum PM$_{2.5}$ emissions, i.e. 15 % of the annual total, followed by Dec–Jan–Feb (10 %) (Fig. S10). The sectoral contribution showed April had peak emissions from agricultural residue burning (51 % of residue

burning), while in Dec–Jan–Feb, the variability in emissions was mostly caused due to temporal variability in space and water heating and agricultural residue burning emissions.

## 4.   Conclusions

A high resolution 1 km × 1 km technology-linked multipollutant emission inventory was developed for

Nepal for the base year 2011. The trends in energy and emissions during the period 2001–2016 were also studied. This work included all the relevant sources of energy-use sectors such as residential, industry, transport, commercial and agriculture (methane from enteric fermentation and manure management from livestock was also estimated), and emissions were characterized by their combustion technologies. These





broad sectors were further disaggregated into sub-sectors of coherent sources and emissions were estimated for a total of 36 combustion technologies. The national energy consumption estimated for 2011 was 378 PJ, with the residential sector being the largest consumer of energy, around 79 %, followed by industry (11 %) and transport (7 %). The energy source was dominated by biomass, contributing 88 % to

the total energy consumption, while 12 % was from fossil fuel. Nationally, 8.4 Tg $CO_2$, 666 Gg $CH_4$, 2.5 Gg $N_2O$, 72 Gg $NO_X$, 1984 Gg CO, 477 Gg NMVOC, 239 Gg $PM_{2.5}$, 28 Gg BC, 99 Gg OC and 28 Gg $SO_2$ was estimated in 2011 using regionally measured emissions factors. The sectoral contributions to emissions were mostly energy-driven, especially from cookstoves using biomass and from open burning in the residential and agriculture sectors. The assessment of top polluting technologies showed high

emissions from traditional cookstoves using firewood, dungcakes, and agricultural residues, and open burning emissions from wood and residues. In addition, high emissions were also computed from fixed chimney Bull's Trench Kilns in brick production, cement kilns, two-wheeler gasoline vehicles, heavy diesel freight vehicles and kerosene lamps. The estimated energy was compared with officially reported energy by WECS for 2011. Overall the technology-linked energy consumption estimated in NEEMI was

higher by a factor 1.15 than estimated by WECS. Comparison with the regional inventory MIX showed lower estimates of $NO_X$, CO, BC, OC and $SO_2$, while $CH_4$, $N_2O$, NMVOC and $PM_{2.5}$ were estimated to be higher, due to differences in methodologies, emission factors and the disaggregation of energy across sources and sectors. The energy and emissions for all of Nepal were also compared with specifically the Kathmandu Valley, and it was found that the Valley consumed 8.2 % of the national energy in 2011 and

the emissions were 3–6% of the national estimates, except 14 % of $NO_X$, which was attributed to a high use of diesel in vehicles and diesel generators.

Activity specific spatial proxies were used for developing the gridded spatial distribution of emissions. A wide variation in emissions distributions was achieved, highlighting the pockets of growing urbanization

and the detailed knowledge about the emission sources. Emissions from brick production, the agriculture sector, space and water heating and the use of diesel generators were apportioned across different months using information on the respective temporal variation in the activities. It was observed that April constituted the maximum $PM_{2.5}$ emissions, followed by Dec–Jan–Feb. Though such extensive work



requires a lot of crucial demographic data and statistical information for each activity, this work has made some pragmatic assumptions whenever required, which identifies the need to improve the information available for some of the important sectors at the national and municipality level. The NEEMI had attempted to provide an insightful and detailed understanding of the energy requirements in various

sectors. It has also provided implications of new energy efficient technologies in emissions, which can be further used in modelling the atmospheric dynamics at urban to regional scale and redesigning appropriate mitigation policies.

## 5.  Author Contribution

MR and ML conceptualized the research idea. PS designed the methodology, investigated and analyzed the results. PB and KS provided inputs on spatial distribution of source sectors and crucial discussion and analysis on emission factors. PS prepared the manuscript with contribution of all co-authors.

## 6.  Acknowledgements

The work is a part of the IASS project "A Sustainable Atmosphere for the Katmandu Valley" (SusKat) funded by the German Federal Ministry for Education and Research (BMBF) and the Brandenburg State Ministry for Science, Research and Culture (MWFK)". The author would like to thank Mr. Bhishma Pandit, for his critical and expert comments on fuel consumption in industry sector; Prof. Rejina Maskey Byanju (Tribhuvan University, Nepal), Mr. Bhupendra Das (doctoral student, Tribhuvan University,

Nepal) and Mr. Pratik Singdan (Research Assistant, IASS Potsdam) for assisting in data collection; Dr. Ashish Singh (Post-doctoral fellow, IASS Potsdam) and Mr. Khadak Mahata (Research Assistant, IASS Potsdam) for fruitful discussions and interpreting the traditional/regional practices in energy consumption across different sectors in Nepal.

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



## List of Figures

## List of Tables



**List of Figures**

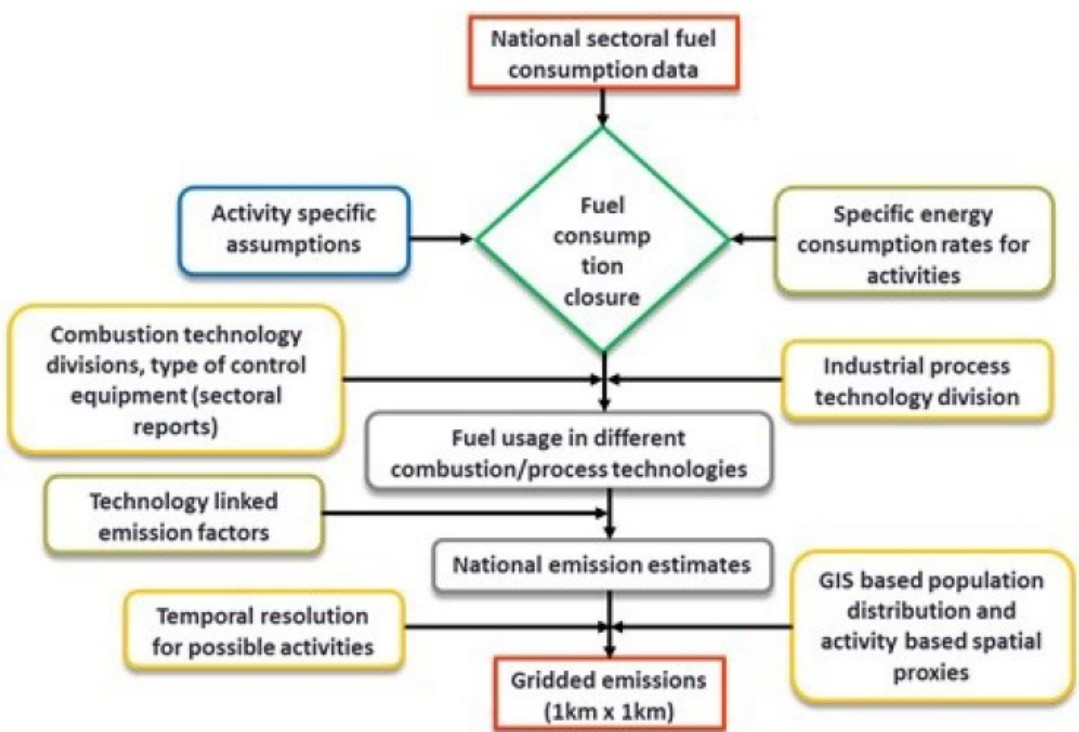

**Figure 1.** Methodology for energy and emissions estimate



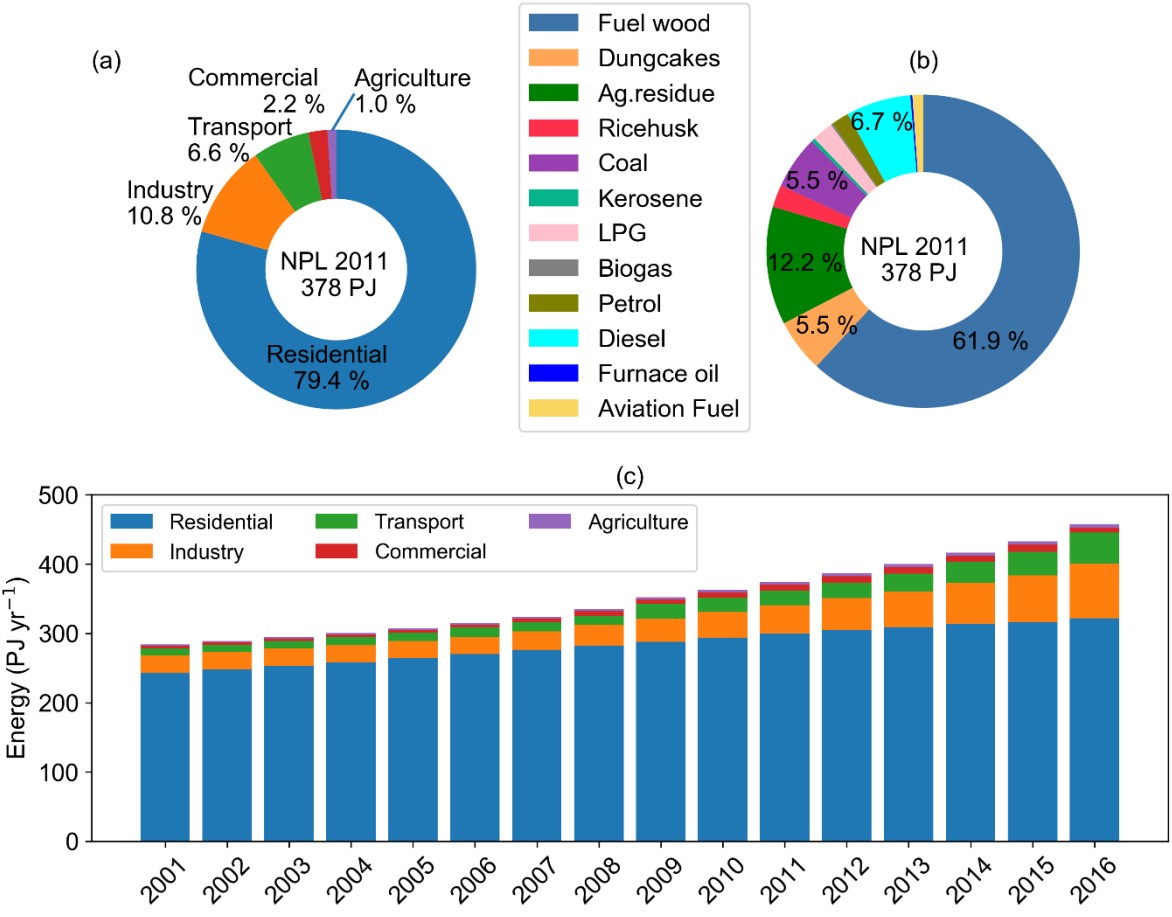

**Figure 2.** (a) National sectoral energy consumption and b) Contribution of fuel type to national energy consumption estimated for 2011 (c) Energy consumption trend for the period 2001-2016



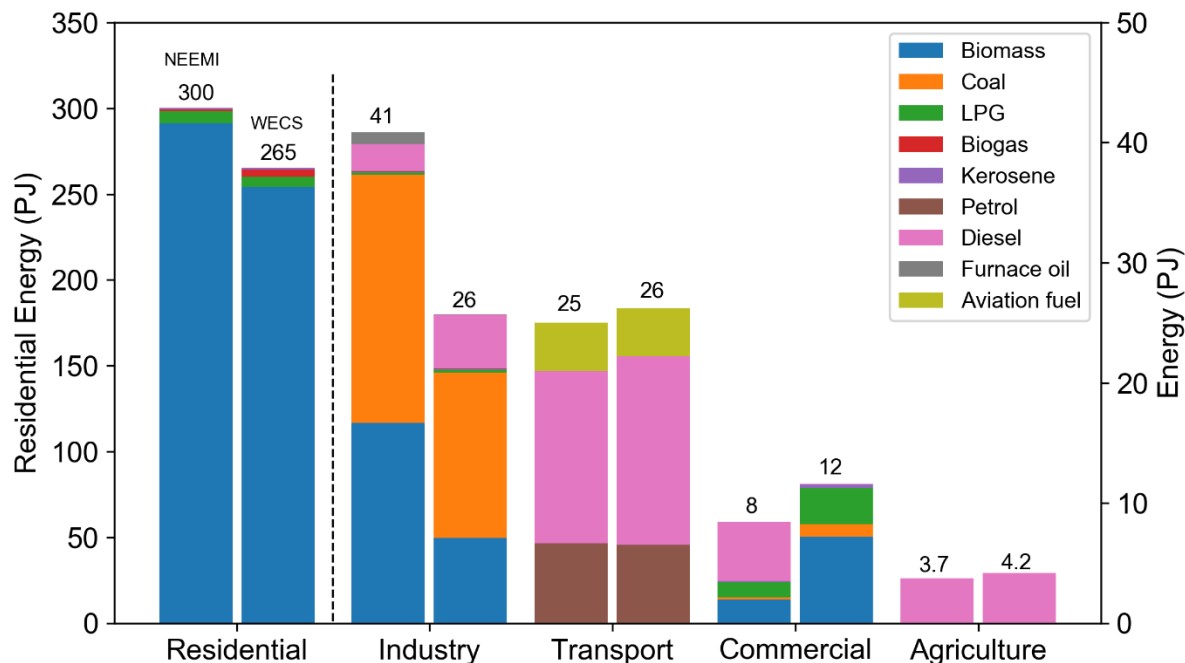

**Figure 3.** National energy consumption estimated for 2011; this study compared with WECS's
estimates. Right Y-axis for Industry, Transport, Commercial and Agriculture sectors, Left Y-axis for
10 Residential sector





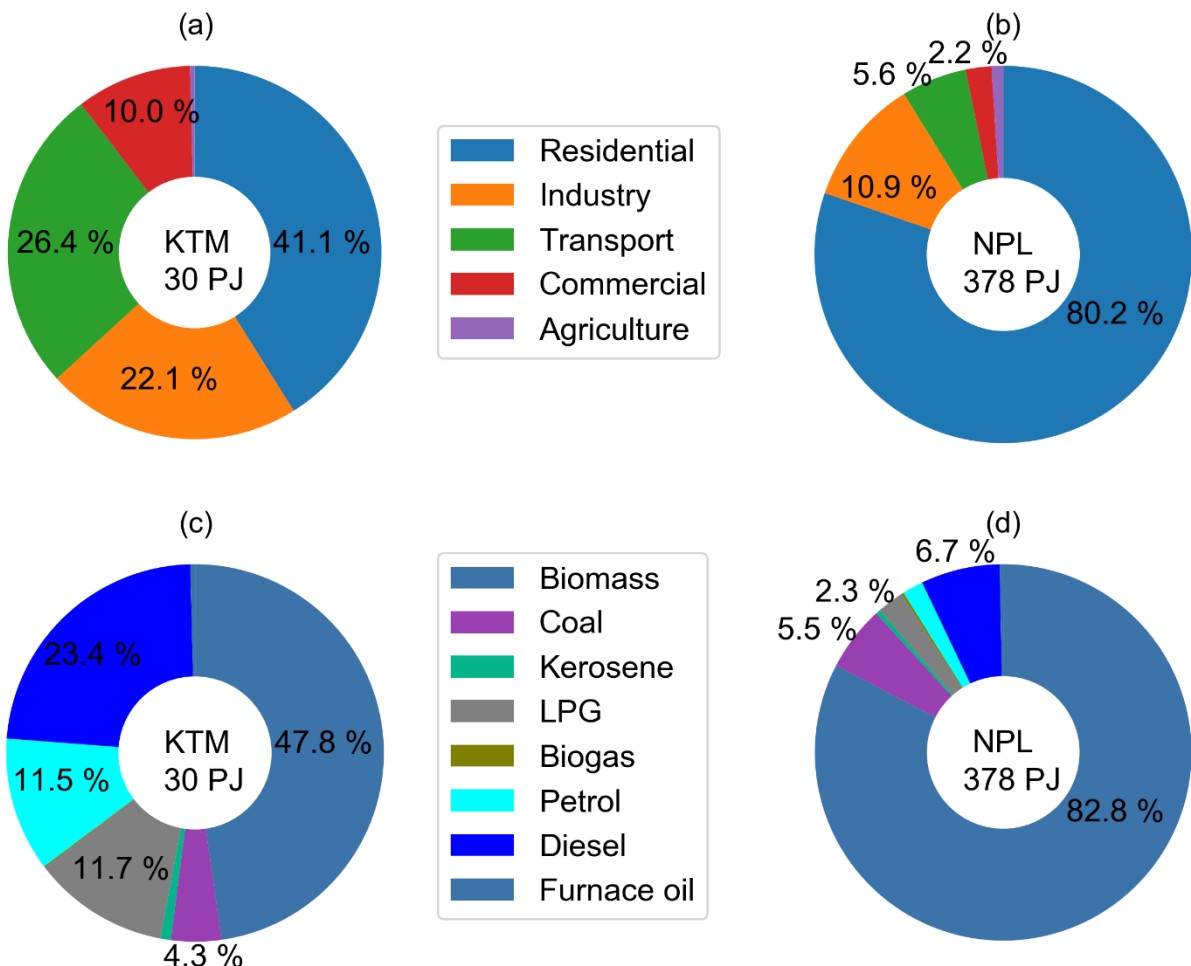

**Figure 4.** Energy comparison at sectoral level for the Kathmandu Valley and Nepal in (a) and (b) and for different fuels (c) and (d) for year 2011





**Figure 5.** National emissions estimates for aerosols and trace gases for (a) each sector in 2011 and (b) normalized trend for a period 2001-2016




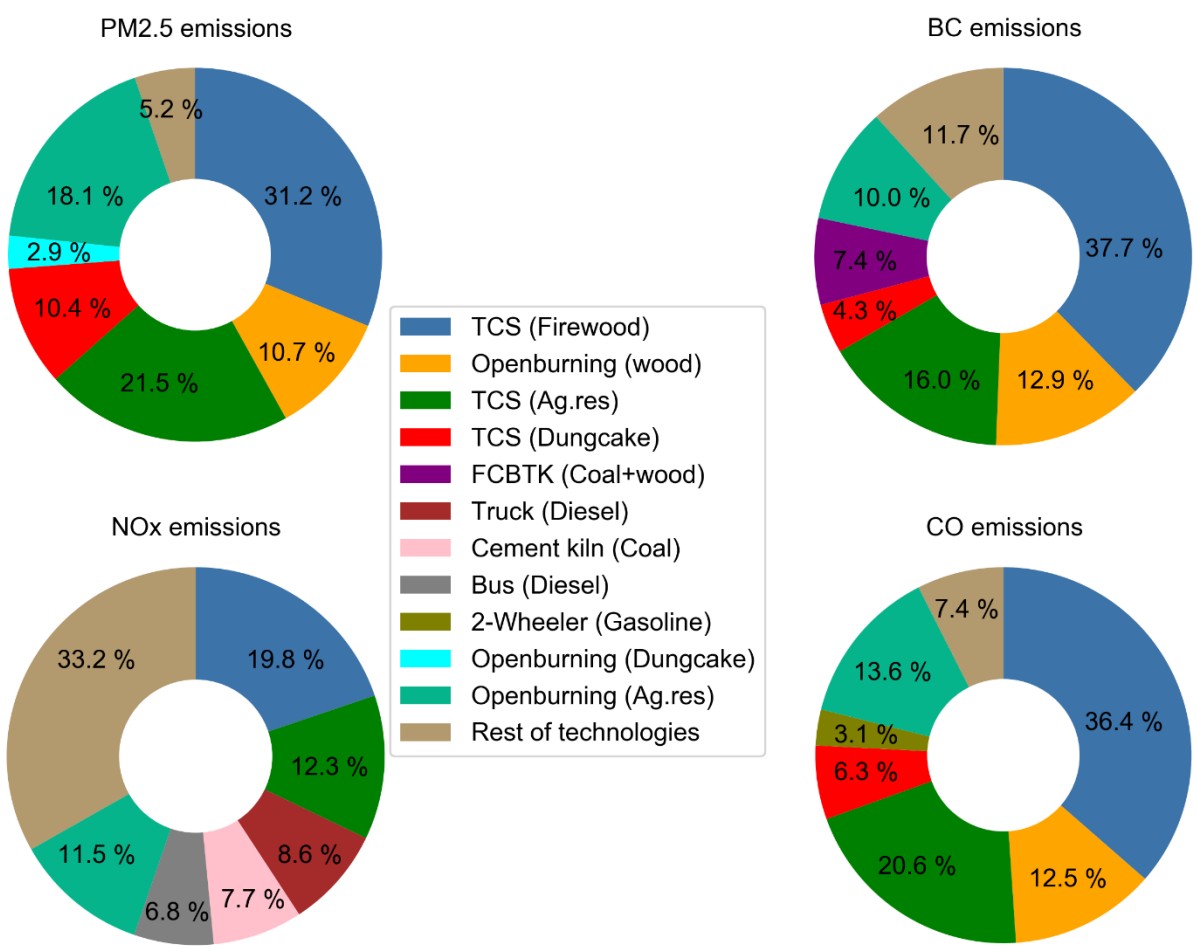

5  **Figure 6.** Top six polluting technologies contributing to (a) PM$_{2.5}$ (b) BC (c) NOx (d) CO emissions





**Figure 7.** Emissions comparison with REAS and MIX HTAP regional inventories





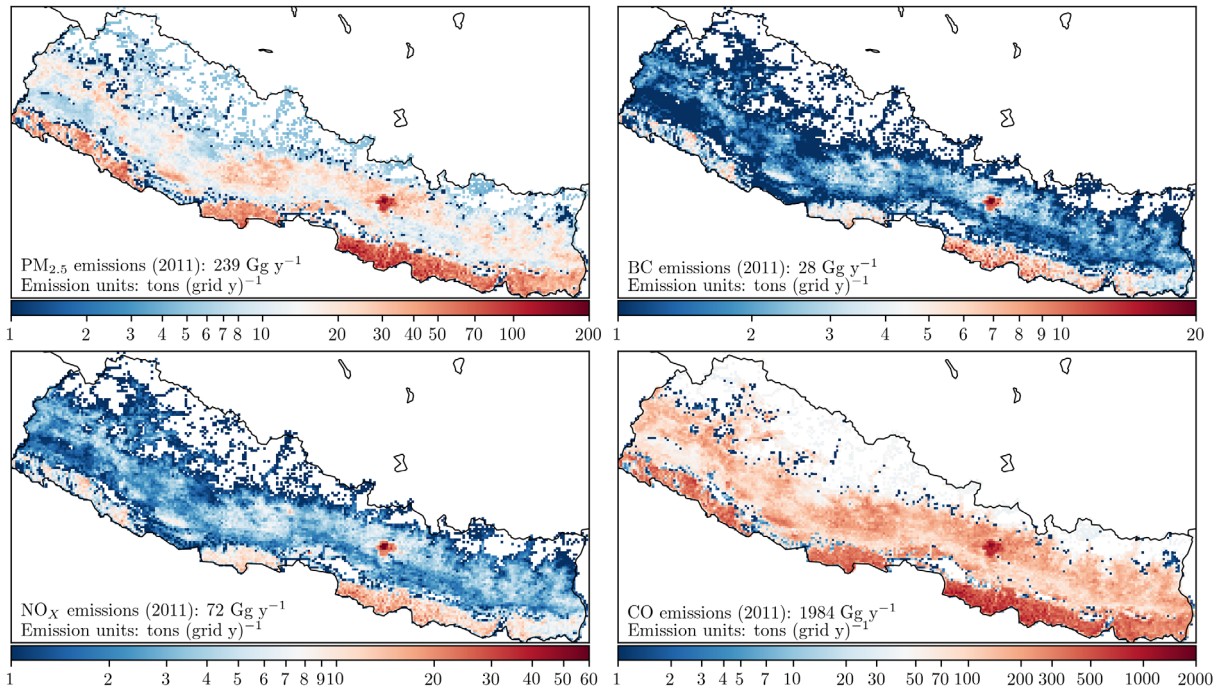

**Figure 8.** Spatial distribution of PM$_{2.5}$, BC, NOx and CO emissions for year 2011

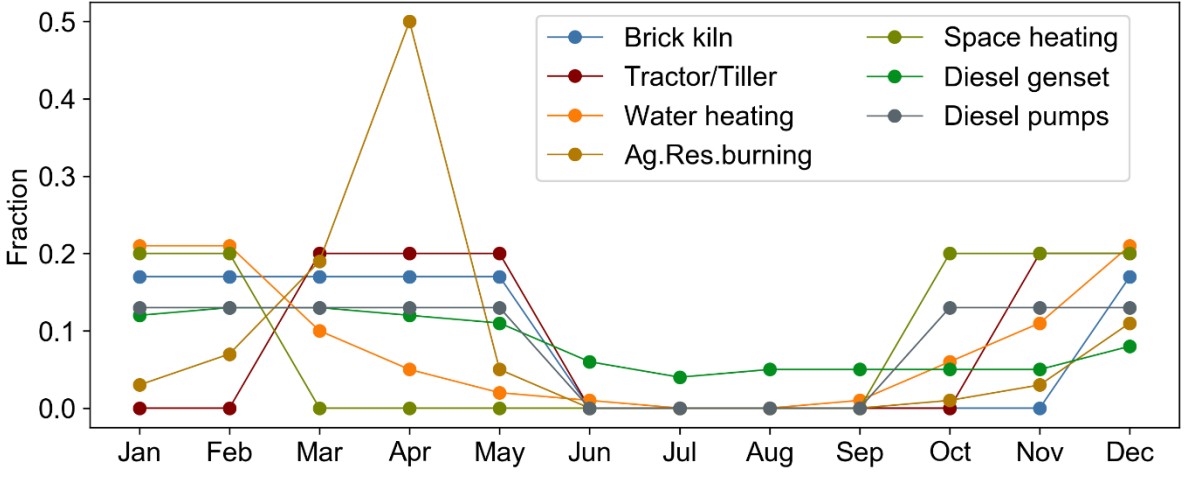

**Figure 9.** Temporal variability accounted in activity rates starting from January till December





## List of Tables

**Table 1.** Details about important features of NEpal EMission Inventory 2011 (NEEMI)

| Features | Details |
|---|---|
| Base year | 2011 |
| Region | 75 districts and urban cities as per Census 2011 |
| Sectors and sub-sectors | *Residential:* Cooking, lighting, space heating, water heating and boiling<br><br>*Industry:* Large point sources of cement, brick production, basic iron, structural metal, pharmaceutical, tea and coffee, grain mill, noodles and other small industries<br><br>*Transport:* two wheelers, cars (gasoline), cars (diesel), bus, micro/mini-bus, mini-trucks, trucks and off-road vehicles<br><br>*Commercial:* Academic institutions, hospitals, financial institutions, government offices, barrack canteens, hotels and restaurants<br><br>*Agriculture:* Diesel pumps, tractors, power tillers and threshers, agricultural residue burning |
| Species | *Aerosols and constituents:* $PM_{2.5}$, BC and OC<br><br>*Ozone precursors and other gases:* $NO_X$, CO, NMVOC and $SO_2$<br><br>*Greenhouse gases:* $CO_2$, $CH_4$ and $N_2O$ |
| Spatial resolution | 1 km × 1 km |
| Temporal resolution | Monthly |



**Table 2.** Parameters used for energy required in water heating and boiling source category

| Parameters | Value | Units |
|---|---|---|
| density of water | 997 | g/l |
| Sp. Heat capacity | 4.1 | J/g-K |
| Average water temperature | 20 | ºC |
| Desired water temperature | 43 | ºC |
| Bathing water requirement | 15 | l/capita-day |

10 **Table 3.** Parameters used for space heating source category

| | No. of hours (hr) | | Fraction household | | Burn rate (kg/hr) | | Days per month | |
|---|---|---|---|---|---|---|---|---|
| | low | high | low | high | low | high | low | high |
| Regular | 3.0 | 5.0 | 0.50 | 0.75 | 0.50 | 1.00 | 15 | 20 |
| At night | 6.0 | 8.0 | 0.10 | 0.20 | 0.50 | 1.00 | 15 | 20 |





**Table 4.** Technology details in each sector and sub-sectors

| Sector | Sub-sector/Activities | Fuel | Combustion & Process technology |
|---|---|---|---|
| RESIDENTIAL | Cooking | Firewood | Traditional mud cookstove (TCS), Improved cookstove (ICS) |
| | | Dungcakes | |
| | | LPG | LPG stove |
| | | Kerosene | Kerosene pressure stove |
| | | Biogas | Biogas stove |
| | Lighting | Kerosene | Kerosene wick lamp |
| | | Biogas | Biogas lamp |
| | Water heating and boiling | Firewood | TCS |
| | | Kerosene | Kerosene pressure stove |
| | | LPG | LPG stove |
| | Space heating | Firewood (In) | TCS |
| | | Firewood (Out) | Open burning |
| | | Dungcakes | Open burning |
| INDUSTRY | Brick kilns | Coal, wood | Fixed Bulls' Trench Kiln, Clamps |
| | Cement production | Coal | Rotary kilns |
| | Basic Iron | Furnace oil | Reheating furnace |
| | Industries | Coal, wood | Furnace |
| | | Ricehusk | Boiler |
| | | Diesel | Diesel generator, oil boiler |
| | | Furnace oil | Oil boiler |



| | | | |
|---|---|---|---|
| **COMMERCIAL** | Academic institutions, government offices hospitals, financial institutions and other service sector | Diesel | Diesel generator |
| | Barrack canteen, Hotel, Restaurants | Coal/Wood | TCS |
| | | Kerosene | Kerosene pressure stove |
| | | LPG | LPG stove |
| | | Diesel | Diesel generator and oil boiler |
| **AGRI-CULTURE** | Agricultural residue burning | Biomass | Open burning |
| | Irrigation Pumps | Diesel/Gasoline | Diesel pump/Gasoline pump |
| | Tractors | Diesel | Diesel tractor |
| | Power tiller | Diesel | Diesel power tillers |
| | Thresher | Diesel | Diesel engines |
| **TRANS-PORT** | Private passenger | Gasoline | Two wheeler, Cars |
| | Public passenger | Diesel | Jeep/Taxi, (Microbus, Minibus), Bus |
| | Public freight | Diesel | (Pick-up, Mini truck), Trucks |
| | Off-road vehicles | Diesel | (Tractors, power tillers, Others) |





**Table 5.** Comparison of NEEMI emissions with the MIX emissions for 2010

| Sectors | $CO_2$ | $CH_4$ | $N_2O$ | $NO_X$ | CO | NMVOC | $PM_{2.5}$ | BC | OC | $SO_2$ |
|---|---|---|---|---|---|---|---|---|---|---|
| NEEMI (2010) | 33.1 | 108.0 | 2.1 | 60.3 | 1683 | 398 | 184 | 24.5 | 96.9 | 14.2 |
| MIX (2010) | 34.0 | 90.0 | 1.5 | 83.0 | 2109 | 377 | 139 | 27.0 | 105.0 | 30.0 |
| Ratio NEEMI/MIX | 0.97 | 1.19 | 1.32 | 0.71 | 0.79 | 1.05 | 1.32 | 0.90 | 0.92 | 0.47 |

*All units in Gg/yr except $CO_2$ in Tg/yr*

5    **Table 6.** Comparison of total and sectoral emissions from the Kathmandu Valley and Nepal for 2011

| Sectors | $CO_2$ | $CH_4$ | $N_2O$ | $NO_X$ | CO | NMVOC | $PM_{2.5}$ | BC | OC | $SO_2$ |
|---|---|---|---|---|---|---|---|---|---|---|
| KTM/NPL | 0.07 | 0.04 | 0.06 | 0.13 | 0.06 | 0.06 | 0.04 | 0.06 | 0.04 | 0.06 |
| Industry | 0.10 | 0.09 | 0.26 | 0.11 | 0.08 | 0.08 | 0.16 | 0.08 | 0.18 | 0.07 |
| Residential | 0.04 | 0.03 | 0.03 | 0.03 | 0.03 | 0.04 | 0.03 | 0.04 | 0.03 | 0.02 |
| Commercial | 0.33 | 0.11 | 0.26 | 0.36 | 0.19 | 0.19 | 0.26 | 0.20 | 0.29 | 0.16 |
| Agriculture | 0.03 | 0.05 | 0.03 | 0.02 | 0.04 | 0.04 | 0.04 | 0.02 | 0.05 | 0.22 |
| Transport | 0.35 | 0.54 | 0.36 | 0.34 | 0.48 | 0.47 | 0.34 | 0.32 | 0.38 | 0.37 |