# Peer review of "Nepal Emissions Inventory - I: Technologies and combustion sources (NEEMI-Tech) for 2001-2016"

_Atmospheric Chemistry and Physics, 2019_

## Referee Comment (RC1) · Anonymous Referee #1 · 30 Apr 2019

A very useful, data rich, and rather well documented paper that deserves publication. However, in my view, it needs some revisions. It would benefit from shortening of the more general sections of the main text while more extensive and especially more focused discussion of results and comparison to other work.

One of the key issues is completeness of the inventory and consequently a decision about the content of the paper. The title indicates it is complete inventory, however, it focuses on anthropogenic and primary combustion sources. If it shall remain as such then the title should be modified and a clear statement about the content should be made already in the abstract. Considering the sources that are currently covered (a

table summarizing the coverage would be a great help) I do not see why the N2O and CH4 is at all included since vast majority of emissions originate from sources NOT included in this work (agriculture, waste). Therefore, I'd suggest to remove these species from the inventory OR complete the dataset with agriculture (livestock, fertilizers, rice) and waste (use and refer to the results of the Das et al, 2018 paper).

Few other comments about the source coverage; - For CO2, it might be useful to specifically show the fossil fuel emissions vs biofuel emissions, possibly giving also a clear information about the assumptions about the non-sustainability of wood supply - it is mentioned but I have not found a clear explanation of assumptions. Since also the CO2 from open biomass burning (agricultural residue) is included in the total it woudl be good to separate it from fossil fuel CO2. - For NMVOC, according to my understanding the solvent use and losses from liquid fuel storage distribution and not included - it would be useful to add a few sentences about the potential contribution of those (it will be rather small I assume but still worth mentioning for completeness and in the future these sources might be gaining in importance). One potentially more important source that might be missing (?) are evaporative emissions from gasoline vehicles - I am not sure if the emission estimates include these in emission factors or not. - The authors mention the Das et al (2018) paper where emissions from waste burning are estimated; I'd suggest to include results of that paper here and quote that for this source emissions come from that paper as these are relevant for all air pollutant species and so the presented inventory would be more complete. - Emissions of SO2 from residential sector are typically not a major source and so it is the case in Nepal but considering the large role solid fuels play in the energy balance the contribution seems very small and I was wondering if the authors should not review the assumptions about the SO2 emission factors used for biomass. - It is not clear if the non-exhaust emissions from transport (brake wear, tyre wear) are included or not.

Comparison to other studies; obviously there is not a lot to compare to but I believe one could add few more sources, like the CMIP6 dataset (Hoesely et al., 2018) where

annual estimates until 2014 were made and are available at sectoral level too, then also the EDGAR inventory (Crippa et al), and GAINS model work where the ECLIPSE project dataset is available (Klimont et al, 2017). The currently presented comparison provided in Fig 7 and one page of discussion comes short (in my view) of what could be discussed as the comparison could serve highlighting, for example key uncertainties or elements that are not well represented or out of date in the regional and global products.

BC and OC emission factors for transport sources. While the Bond et al (2004) work certainly provides a great starting point the shares presented there referred submicron PM so one needs to carefully use those numbers (for transport where exhaust PM emissions are primarily in submicron fraction this is probably fine). However, there has been a lot more work done assessing diesel carbonaceous emissions, including work in Asia and also how the BC share changes in vehicles with more advanced emission standards - something that seems to be ignored here; please review the possibility of using also some more recent studies and consider the extent to which for more recent vehicles the factors might be different. It is not clear form the current text, if the data about the share of fuel used by vehicles complying with different Bharat standards is available and explicitly used or not. There has been few more studies looking at high emitters and the implications for emissions. It might be useful to add few more lines of discussion and in the result section highlight the importance of this source for specific pollutants - even show this in the charts explicitly as it offers a good policy target.

Continuing on the above, but for stationary sources, the use of BC and OC shares of PM2.5 where the abatement technologies, like particulate filters, are not considered is problematic since application of such filters changes the size profile (not reducing emissions of all PM species equally). Even if such information is not available as yet, it should be mentioned and for power plants and industrial boilers the installation of such stack control technology will become reality sooner or later.

The aviation sector - The autors state in line 23-24 on page 11 that it is not included

but in several parts of the manuscript and also all energy graphs aviation is mentioned. It should be made consistent.

On page 15, line 23 - one of the paper has 'way to high' emisison factors....it woudl be good to add some more justification to why this study is excluded ...or maybe shoudl not be mentioned at all if it is wrong.

Section 3.1.1 mentions also burning of ag residues but i would not include this under 'national energy trend"

---

## Referee Comment (RC2) · Anonymous Referee #2 · 8 May 2019

General comments Overall, this is an interesting, well-researched piece of work and is worthy of publication. However, there is some confusion about the scope of the study. Although the abstract states that the study focuses on the energy-use sectors and agro-residue open-burning, Section 3.2 also includes estimates of agricultural CH4 emissions from livestock (but not rice cultivation) although these estimates do not appear in any graphs or tables as far as I can see. This is rather confusing and perhaps CH4 should be omitted altogether (as Referee #1 suggested). However, if CH4 is to be included, then perhaps all other major anthropogenic CH4 sources should be covered as well, including rice paddy, landfill waste, waste-water etc. Also, if the study were to be expanded to include non-combustion agricultural activities, then adding ammonia

(NH3) (e.g. from livestock manure management and application of N-fertilizers) to the list of inventoried pollutants would also add value. This is because NH3 is a very important precursor (in addition to NOX and SO2) of secondary inorganic aerosol, and so would need to be included if the inventory results were ever to be used as input to subsequent atmospheric chemistry transport modelling. I also agree with Referee #1 that comparisons of the results with other inventory initiatives that cover Nepal, especially EDGAR and GAINS-Eclipse, would enhance this study.

Specific comments Scope of study. On page 22 (lines 1 and 2) it states that 'For analysis and comparison purposes, only the combustion based emissions from energy sources are considered, leaving out fugitive emissions from livestock management.' So this begs the question: What about open-burning of crop residues – were these included? Residential open burning. Not clear what is meant by residential 'open burning emissions of wood and residues' on line 20 of Page 2, 'heating outside' on lines 17-18 of page 8 and 'Space heating, Open burning' in Table S2. If this is this just burning of biomass in open fires indoors this should be clarified. The term 'open burning' suggest to me that the fires are located outside the house (in the open) which seems a strange way to heat a house. Page 5, line 6: Does the 15-fold increase in vehicle registrations over 2 decades equate to an actual 15-fold increase in vehicle numbers – or just better enforcement of registration rules? Line 16: Should this be 'IPCC Tier 1' or 'EMEP/EEA Tier 1' or both? Kerosene lamps. Little distinction is made between kerosene wick lamps and kerosene hurricane lamps – (although 'kerosene lanterns' are referred to once on page 22, line 9, which I assume equates to hurricane lamps?). In Table S2, for BC, only the emission factor (EF) for wick lamps (90 g/kg) is given, from Lam et al (2012), although that paper (Table S5) estimates 20% of kerosene used for lighting in this region is likely to be in hurricane lamps (EF for BC is 9 g/kg). Do the authors therefore assume no hurricane lamp use for their calculations? Also, in Table S2, Lam et al (2012) is given as the source of the N2O and SO2 EFs, but these do not exist in that paper (as far as I can see) and the OC EF of 0.52 g/kg in Table S2 compares with the average of 0.4 g/kg for wick lamps given in Lam et al. Please

could the authors correct and/or explain the derivation of their EFs for kerosene-fuelled lighting. Brick kilns: Page 13, line 2: It would be nice to know how many of the 557 FCBTKs had the zig-zag firing technology, and also how many VSBK there were.

Technical corrections Page 2, line 14: 'Tons' is not and SI unit – presumably this should be 'tonnes? Then Gg is used thereafter. Consistency in use of units required – suggest using Gg throughout (or Tg for $CO_2$ and CO). Page 4, line 21: Should 'intake' be 'exposure'? Line 22: children (not childrens') Page 6, line 6: 'arising' not 'arousing'. Line 16, Delete '-forcing' as SLCPs stand for 'short-lived climate pollutants'. Page 9, line 7: Replace 'rest' with 'the remaining'. Line 8, 'LPI' and 'SMI' should have been defined earlier in this paragraph. Line 26: Insert 'data on' between 'provided' and 'how'. Page 11, line 22: Should be 'we intend' not 'we tend'. Page 13, line 19: Who is the personal (not personnel) communication from? Page 17, line 7: Replace 'they' with 'there'. Page 19, line 2: Replace 'small increase in' with 'slightly higher level of'. Line 6, Replace 'increase' with 'difference'. Line 8, insert 'being' at start of line. Page 21, line 18: Replace 'spike in 2016 energy' with 'large increase in 2016 energy use'. Page 24, line 1: For 'aerosols' the %$SO_2$ is included in the list yet NOx is also an important precursor of secondary inorganic aerosol – why not include this too? If $NH_3$ were to be added to the inventory (see general comments), this would also need to be added for the same reason. Page 24, line 2: High OC emissions are referred to as shown in Fig 6 – but CO is in that fig, not OC. Has there been a mix-up over OC versus CO? Page 22, line 19: The text includes the 2011 emission estimate for CO and then on page 23, lines 5 & 6, emissions of $CO_2$, $CH_4$ and $N_2O$. Why were these not included in Figure 5(a) or perhaps in a separate table? Page 25, lines 19 & 20: Values given here for all species apart from $SO_2$ are slightly different from those given in Table 5. Page 27, lines 23& 24: Need to rephrase this – I don't think diesel gen-sets were changed to zig-zag firing! Figure 9: Make clear this graph is for 2011. Table 6: Make clear that these values are emission ratios (MTM/NPL). Table S3: Footnote 'e' refers to the liquid fuel combustion in industry having a 22.5% sulfur retention. But liquid fuels leave no ash and so there should be zero sulfur retention in ash – so this must be wrong. If

this footnote should have applied to coal use in industry, then again 22.5% looks wrong as USEPA's AP42 (5th edition, Section 1.1.3.2) implies only 5% retention-in-ash for bituminous coal (the type of coal used in Nepal). Please explain.
* * *

---

## Author Comment (AC1) · 12 Jul 2019

**RESPONSE TO REVIEWERS' COMMENTS**

**Title: "Nepal Emission Inventory (NEEMI): A high resolution technology-based bottom-up emissions inventory for Nepal 2001-2016"**

Pankaj Sadavarte*[a,c], Maheswar Rupakheti*[a], Prakash V. Bhave[b], Kiran Shakya[b], Mark G. Lawrence[a]

[a]Institute for Advanced Sustainability Studies (IASS), Berliner Str. 130, 14467 Potsdam, Germany
[b]International Centre for Integrated Mountain Development (ICIMOD), Lalitpur, Nepal
[c]Now at SRON Netherlands Institute for Space Research, Utrecht, The Netherlands

*Corresponding Authors: Pankaj Sadavarte (p.sadavarte@sron.nl) and Maheswar Rupakheti (Maheswar.rupakheti@iass-potsdam.de)

We thank both referees for their constructive comments. The scientific and grammatical comments have been carefully considered and addressed through the changes detailed below. The comments from both referees have been broken down into points and subsequently answered. A point by point response is included indicating where changes appear in the revised manuscript along with the subsequent changes. The referee's comments appear in black and response to the comments follows immediately after in blue color.

**Note:**
1. All the changes in the revised manuscript are highlighted in blue color.
2. Line numbers indicated in this text refer to the revised manuscript.

**ANONYMOUS REFEREE #1**

1) A very useful, data rich, and rather well documented paper that deserves publication. However, in my view, it would benefit from shortening of the more general sections of the main text while more extensive and especially more focused discussion of results and comparison to other work.

Response: Thank you for a positive note on our study. The section (2.1) Activity rates and technology division, (2.2) Combustion technologies and (2.3) Emission factors are trimmed down to more concentrated information and additional details are moved to the Supplementary Information as section S1, S2, and S3, respectively. We have also provided a more extensive comparison of our emission estimates with five other available global and regional emissions data sets, and the results are discussed extensively.

2) One of the key issues is completeness of the inventory and consequently a decision about the content of the paper. The title indicates it is complete inventory, however, it focuses on anthropogenic and primary combustion sources. If it shall remain as such then the title should be modified and a clear statement about the content should be made already in the abstract.

Response: We agree with the reviewer that we have mixed technology-based emissions and open burning (and fugitive) emissions. Given the extensive nature of completing the open burning and fugitive emissions component, and in order to present the emissions for these two broad sectors more distinctly, we would like to split the manuscript into two parts, pending approval by the editor, as follows:

*Nepal Emission Inventory - I: A high resolution bottom-up combustion and technology-based emissions inventory (NEEMI-Tech) for 2001-2016*

*Nepal Emission Inventory - II: A high resolution bottom-up open burning and fugitive emissions inventory (NEEMI-open) for 2001-2016*

The title of the paper (part 1) has been modified accordingly. The abstract now mentions sectors, clearly highlighting that the current manuscript presents only combustion and technology-based emissions. All the sections and sub-sections on open burning emissions are now removed from part 1. We will prepare part 2 for publication as soon as possible.

A paragraph has been inserted clearly mentioning the scope of the study, sectors and pollutants on page 6 line 16 of the revised manuscript:

*Analyzing the following issues, it is important to conduct a systematic and comprehensive study of all energy sectors, agriculture sources and solid waste burning in Nepal from an emissions point of view, which has not yet been done, integrating the primary information on energy production and use, fuel combustion technologies and corresponding EFs. The Nepal emissions inventory study is divided into two parts; technology-based emissions (NEEMI-Tech) as part I, and open burning and fugitive emissions (NEEMI-Open) as part II. This paper discusses the development of a high resolution (1 km × 1 km, monthly) combustion and technology-based emission inventory from the residential, industrial, transport (on-road and off-road), and commercial sectors, as well as the agricultural sector (only technology-based emissions from use of tractors, tillers, pumps and threshers), while part-II encompasses emissions from open burning of municipal wastes, agricultural open field burning, and forest fires, along with fugitive emissions from waste, paddy fields, enteric fermentation and manure management. Part -II is under preparation for publication. In both parts, emissions of a total of ten species, where applicable, are estimated in this study, including greenhouse gases and short-lived climate-forcing pollutants (SLCPs).*

3) Considering the sources that are currently covered (a table summarizing the coverage would be a great help)

Response: A Table with details like the source, pollutants, spatial and temporal coverage, and base year of the inventory is presented as "Table 1. Details about important features of NEpal EMission Inventory (NEEMI)."

4) I do not see why the N2O and CH4 is at all included since vast majority of emissions originate from sources NOT included in this work (agriculture, waste). Therefore, I'd suggest to remove these species from the inventory OR complete the dataset with agriculture (livestock, fertilizers, rice) and waste (use and refer to the results of the Das et al, 2018 paper).

Response: We had originally included $N_2O$ and $CH_4$ emissions with the intention to estimate the burden of all the important pollutants arising from the combustion-based sources in Nepal. However, the referee is quite correct that for these two gases, the sources covered in this part of the database are very minor compared to their other sources, and we will save describing those major source of emissions until the second part is ready (having now split the complete study in two parts as explained above). The revised manuscript now contains only technology-based combustion emissions.

5) Few other comments about the source coverage; - For CO2, it might be useful to specifically show the fossil fuel emissions vs biofuel emissions, possibly giving also a clear information about the assumptions about the non-sustainability of wood supply - it is mentioned but I have not found a clear explanation of assumptions. Since also the CO2 from open biomass burning (agricultural residue) is included in the total it would be good to separate it from fossil fuel CO2.

Response: The following analysis describing the difference in CO2 emissions from biofuel and fossil fuel is added on page 21 line 7:

*Broadly, fossil fuels and biofuels (firewood, agriculture residue, dungcakes and biogas) contribute 73 % and 27 %, respectively, to CO2 emissions in 2011, in contrast to only 12 % (43 PJ) of the energy coming from fossil fuels, and 88 % (331 PJ) from biofuels, since fossil fuel combustion moves CO2 from the long-term fossil reservoir into the atmosphere, while biofuels are mostly recycling of recent biomass (though they include some long-term removal of net biomass, which is small relative source to the atmosphere). Sectorally, the industry sector contributes 46 % of the combustion-based national CO2 emissions, followed by residential (32 %) and transport (15 %), with small fractions from the commercial (4 %) and agricultural (mechanized farming) (3 %) sectors. Even though the residential activities are the main drivers of the national energy consumption, consuming 292 PJ of biofuels and 8 PJ of fossil energy, almost 92 % of the CO2 emissions from biofuels are from recycled carbon, with the non-recycled fraction being about 10 % each from firewood, agricultural residue and 0 % dung cakes, according to Venkataraman et al. (2010). The CO2 emissions from the industries are mostly driven by fossil fuel combustion (98 % fossil fuel and 2 % biofuel), especially coal use in the production of bricks (27 %) and process emissions (53 %) from cement manufacturing.*

6) For NMVOC, according to my understanding the solvent use and losses from liquid fuel storage distribution and not included - it would be useful to add a few sentences about the potential contribution of those (it will be rather small I assume but still worth

mentioning for completeness and in the future these sources might be gaining in importance).

Response: Since we have now decided to include only combustion and technology-based emissions in part I, any fugitive emissions such as solvent use and evaporative loss will be provided in part-II. The following brief analysis on fugitive emissions of NMVOC is added on page 21 line 4:

*The fugitives of NMVOC from storage tanks and service stations, and the transport and evaporative emissions from gasoline vehicles, which are expected to be a small fraction compared with combustion-related NMVOC emissions (though they might gain importance in the future) will be considered in detail in the second part of this study.*

7) One potentially more important source that might be missing (?) are evaporative emissions from gasoline vehicles - I am not sure if the emission estimates include these in emission factors or not.

Response: See response in comment #6.

8) The authors mention the Das et al (2018) paper where emissions from waste burning are estimated; I'd suggest to include results of that paper here and quote that for this source emissions come from that paper as these are relevant for all air pollutant species and so the presented inventory would be more complete.

Response: As mentioned earlier in point 4, now we have split the NEEMI study in two parts as explained in detail on page 6 line 16. Please see the response to comment #2 to see the new paragraph inserted in the text.

9) Emissions of SO2 from residential sector are typically not a major source and so it is the case in Nepal but considering the large role solid fuels play in the energy balance the contribution seems very small and I was wondering if the authors should not review the assumptions about the SO2 emission factors used for biomass.

Response: Thank you for the suggestion. We have now revised the emission factors for $SO_2$ from firewood burning. Earlier, the $SO_2$ EF value of 0.08 g/kg from Habib et al. (2004) was used. Now, additional papers (Saud et al., 2011, Stockwell et al., 2015) were reviewed and a new EF has been used, which was obtained by averaging EFs from Habib et al. (2004), Saud et al. (2011) and Stockwell et al. (2015), i.e., (0.08+0.26+0.499)/3 = 0.28 g/kg. The change has been incorporated throughout the analysis and updated EFs can be seen in Table S1.

10) It is not clear if the non-exhaust emissions from transport (brake wear, tyre wear) are included or not.

Response: Non-exhaust emissions are not included. We inserted the following statement in the transport EF section on page 13 line 20:

*Non-exhaust emissions such as brake wear and tire wear are not included in this study.*

11) Comparison to other studies; obviously there is not a lot to compare to but I believe one could add few more sources, like the CMIP6 dataset (Hoesely et al., 2018) where annual estimates until 2014 were made and are available at sectoral level too, then also the EDGAR inventory (Crippa et al), and GAINS model work where the ECLIPSE project dataset is available (Klimont et al, 2017). The currently presented comparison provided in Fig 7 and one page of discussion comes short (in my view) of what could be discussed as the comparison could serve highlighting, for example key uncertainties or elements that are not well represented or out of date in the regional and global products.

Response: Thank you for pointing to several global and regional inventories. We have now compared NEEMI-Tech emissions with technology-based emissions from the EDGAR, CMIP6, ECLIPSE-GAINS, REAS and MIX HTAP emission inventories. The comparisons are now included in Figure 7 and a new text has been inserted in the manuscript page 24, line 26 discussing the comparison results:

[revised manuscript text omitted]

12) BC and OC emission factors for transport sources. While the Bond et al (2004) work certainly provides a great starting point the shares presented there referred submicron PM so one needs to carefully use those numbers (for transport where exhaust PM emissions

are primarily in submicron fraction this is probably fine). However, there has been a lot more work done assessing diesel carbonaceous emissions, including work in Asia and also how the BC share changes in vehicles with more advanced emission standards - something that seems to be ignored here; please review the possibility of using also some more recent studies and consider the extent to which for more recent vehicles the factors might be different.

Response: Emission factors for BC and OC are revised and updated from more recent work on diesel and gasoline vehicles. The description about BC and OC fraction is added on page 13 line 12:

*The fractions of BC and OC were obtained by averaging respective fractions from chassis dynamometer test results by Kim Oanh et al. (2010), Wu et al. (2015), Zhang et al. (2015), Yang et al. (2019) and Jaiprakash et al. (2016). These studies reflect the regional characteristics of driving cycles and the road infrastructure, which plays an important role in tail pipe exhaust. Kim Oanh et al., (2010) made a vintage based measurements showing the degradation of emissions and resultant high fractions of EC and OC in the oldest category of vehicles. However, Wu et al., (2015) and Jaiprakash et al., (2016) emphasized the importance of driving speed (on non-highways, highways and in cities) on the EC and OC fraction from diesel vehicles.*

13) It is not clear form the current text, if the data about the share of fuel used by vehicles complying with different Bharat standards is available and explicitly used or not.

Response: The share of fuel used by vehicles complying with different Bharat Standards is explicitly used and mentioned while estimating the $SO_2$ emissions during 2001-2016 on page 13, line 19:

*SO2 emissions were calculated using the sulfur content of BS-II/III/IV fuel imported from India, with no retention assumed.*

14) There has been few more studies looking at high emitters and the implications for emissions. It might be useful to add few more lines of discussion and in the result section highlight the importance of this source for specific pollutants - even show this in the charts explicitly as it offers a good policy target.

Response: An analysis on superemitters is added in the manuscript on page 20 line 5:

*A 40 % of the fleet is assumed to be superemitters or high emitters and they contribute to 53 % PM2.5, 58 % BC and 44 % OC emissions from the transport sector. Their emission factors were derived by scaling the normal emission factors (Klimont et al., 2017). The scaling factor was derived from Klimont et al., (2017), who provided the global scaling factors for diesel and gasoline vehicles across different emission standard vehicles, and also based on the measurements of emission factors for PM (Subramanian et al., 2009) in developing countries. On an average, light duty diesel vehicles are scaled by a factor 5 to get a value of 8.1 g/kg and an average heavy duty diesel vehicles are scaled by a factor 2 to get a value of 13.3 g/kg for superemitter vehicles. Though, there is no clear*

*distinction and measured emission factors for a superemitter vehicle, many contemporary studies have derived it using statistical percentile (Subramanian et al., 2009, Ban-Weiss et al., 2009). If a policy to identify and remove the superemitters is enforced, that would reduce ~30 % of PM2.5, BC and OC transport sector emissions, which can be considered as a good and immediate policy target.*

15) Continuing on the above, but for stationary sources, the use of BC and OC shares of PM2.5 where the abatement technologies, like particulate filters, are not considered is problematic since application of such filters changes the size profile (not reducing emissions of all PM species equally). Even if such information is not available as yet, it should be mentioned and for power plants and industrial boilers the installation of such stack control technology will become reality sooner or later.

Response: We have added the following sentence on page 22 line 23:

*The information on emission control measures used in the industries is not readily available. Given poor implementation of policies in Nepal on emission control measures, we expect that most of the industries are operated without proper emission control measures. However, the installation of stack emission control technology in the power plants and industrial boilers will become a reality sooner or later. This is likely to change the emission profile, such as shares of BC and OC in PM2.5, and the size of particles emitted, which needs to be considered in future emission estimates.*

16) The aviation sector - The autors state in line 23-24 on page 11 that it is not included but in several parts of the manuscript and also all energy graphs aviation is mentioned. It should be made consistent.

Response: We have completely removed the mention of aviation energy from the manuscript. The transport sector, henceforth, only consists of on-road and off-road vehicular energy and emissions.

17) On page 15, line 23 - one of the paper has 'way to high' emission factors....it would be good to add some more justification to why this study is excluded ...or maybe should not be mentioned at all if it is wrong.

Response: The phrase 'way to high' relating to the emission factors has been removed.

18) Section 3.1.1 mentions also burning of ag residues but i would not include this under 'national energy trend"

Response: Agricultural residue burning on fields is completely removed from this part of the inventory paper and is being considered in part-II, i.e., NEEMI-Open. Details are mentioned in page 6 line 16. Please see the new paragraph in the response to comment #2.

---

## Author Comment (AC2) · 12 Jul 2019

**RESPONSE TO REVIEWERS' COMMENTS**

**Title: "Nepal Emission Inventory (NEEMI): A high resolution technology-based bottom-up emissions inventory for Nepal 2001-2016"**

Pankaj Sadavarte*[a,c], Maheswar Rupakheti*[a], Prakash V. Bhave[b], Kiran Shakya[b], Mark G. Lawrence[a]

[a]Institute for Advanced Sustainability Studies (IASS), Berliner Str. 130, 14467 Potsdam, Germany
[b]International Centre for Integrated Mountain Development (ICIMOD), Lalitpur, Nepal
[c]Now at SRON Netherlands Institute for Space Research, Utrecht, The Netherlands

*Corresponding Authors: Pankaj Sadavarte (p.sadavarte@sron.nl) and Maheswar Rupakheti (Maheswar.rupakheti@iass-potsdam.de)

We thank both referees for their constructive comments. The scientific and grammatical comments have been carefully considered and addressed through the changes detailed below. The comments from both referees have been broken down into points and subsequently answered. A point by point response is included indicating where changes appear in the revised manuscript along with the subsequent changes. The referee's comments appear in black and response to the comments follows immediately after in blue color.

**Note:**
1. All the changes in the revised manuscript are highlighted in blue color.
2. Line numbers indicated in this text refer to the revised manuscript.

**ANONYMOUS REFEREE #2**

1) General comments Overall, this is an interesting, well-researched piece of work and is worthy of publication. However, there is some confusion about the scope of the study. Although the abstract states that the study focuses on the energy-use sectors and agro-residue open-burning, Section 3.2 also includes estimates of agricultural CH4 emissions from livestock (but not rice cultivation) although these estimates do not appear in any graphs or tables as far as I can see. This is rather confusing and perhaps CH4 should be omitted altogether (as Referee #1 suggested). However, if CH4 is to be included, then perhaps all other major anthropogenic CH4 sources should be covered as well, including rice paddy, landfill waste, waste-water etc.

Response: Thank you for the positive note on our study and for the comment on open burning of agro-residue and CH4 emission from agricultural sources. We have now split the manuscript into two parts, as mentioned in the response to comments by the reviewer no. 1: part 1 on technology-related emissions and part 2 on open burning and fugitive emissions. The above mentioned discrepancies are now rectified and subsequent changes are made in the abstract on page 2 line 6 and kept consistent throughout the manuscript.

*We estimate emissions of aerosols, trace gases and greenhouse gases from five energy-use sectors of residential, industry, commercial, agriculture (only use of tractors, tillers,*

*pumps and threshers) and transport (on-road and off-road) for the period 2001–2016 (with 2011 as the base year), using bottom-up methodologies.*

The scope of the study, sectors and pollutants are clearly mentioned on page 6 line 16:

*Analyzing the following issues, it is important to conduct a systematic and comprehensive study of all energy sectors, agriculture sources and solid waste burning in Nepal from an emissions point of view, which has not yet been done, integrating the primary information on energy production and use, fuel combustion technologies and corresponding EFs. The Nepal emissions inventory study is divided into two parts; technology-based emissions (NEEMI-Tech) as part I, and open burning and fugitive emissions (NEEMI-Open) as part II. This paper discusses the development of a high resolution (1 km × 1 km, monthly) combustion and technology-based emission inventory from the residential, industrial, transport (on-road and off-road), and commercial sectors, as well as the agricultural sector (only technology-based emissions from use of tractors, tillers, pumps and threshers), while part-II encompasses emissions from open burning of municipal wastes, agricultural open field burning, and forest fires, along with fugitive emissions from waste, paddy fields, enteric fermentation and manure management. Part -II is under preparation for publication.  In both parts, emissions of a total of ten species, where applicable, are estimated in this study, including greenhouse gases and short-lived climate-forcing pollutants (SLCPs).*

2)  Also, if the study were to be expanded to include non-combustion agricultural activities, then adding ammonia (NH3) (e.g. from livestock manure management and application of N-fertilizers) to the list of inventoried pollutants would also add value. This is because NH3 is a very important precursor (in addition to NOX and SO2) of secondary inorganic aerosol, and so would need to be included if the inventory results were ever to be used as input to subsequent atmospheric chemistry transport modelling.

Response: We agree with the reviewer that NH3 and SO2 are important species for secondary inorganic formation. Now, since we have split the manuscript into two parts, NH3 emissions from agriculture will be discussed in part 2 of the study as mentioned on page 6 line 16 ( see the inserted paragraph in the response to comment # 1)

3)  I also agree with Referee #1 that comparisons of the results with other inventory initiatives that cover Nepal, especially EDGAR and GAINS-Eclipse, would enhance this study.

Response: Thank you for pointing to several global and regional inventories. We have now compared NEEMI-Tech emissions with technology-based emissions from the EDGAR, CMIP6, ECLIPSE-GAINS, REAS and MIX HTAP emission inventories. The comparisons are now included in Figure 7 and a new text has been inserted in the manuscript page 24 line 26:

*The emissions were also compared… about sulfur retention in ash.*

4)  Specific comments Scope of study. On page 22 (lines 1 and 2) it states that 'For analysis and comparison purposes, only the combustion based emissions from energy sources are considered, leaving out fugitive emissions from livestock management.' So this begs the question: What about open-burning of crop residues – were these included?

Response: The above confusing sentence has been deleted. Now, the scope of the study, sectors and pollutants are clearly specified in the manuscript on page 6 line 16. No emissions from agricultural residue burning (open field) are included in the revised manuscript.

5)  Residential open burning. Not clear what is meant by residential 'open burning emissions of wood and residues' on line 20 of Page 2, 'heating outside' on lines 17-18 of page 8 and 'Space heating, Open burning' in Table S2. If this is this just burning of biomass in open fires indoors this should be clarified. The term 'open burning' suggest to me that the fires are located outside the house (in the open) which seems a strange way to heat a house.

Response: The inconsistency in describing the activity has been corrected and made consistent on the above mentioned lines. For example,
(i) 'open burning emissions of wood and residues' is changed to '*space heating', on page 2 line 19*
(ii) 'heating outside' is expressed in a more convincing way to '*space heating outdoors means, where people gather around an open fire to keep themselves and the immediate vicinity from cold by burning firewood, agricultural residue and dungcakes.' On page 8 line 19*
(iii) 'Space heating, Open burning' is changed to '*Space heating, burning fuel (indoor and outdoor)' in SI Table S1*

6)  Page 5, line 6: Does the 15-fold increase in vehicle registrations over 2 decades equate to an actual 15-fold increase in vehicle numbers – or just better enforcement of registration rules?

Response: The 15–fold increase in vehicle registrations over 2 decades roughly equate to a 15-fold increase in vehicle numbers because there was no vehicle retirement policy in place until only recently, i.e., until March 2017 (e.g, phasing out private transport vehicles older than 20 years). This resulted in virtually all registered vehicles running on the streets. However, in order to account for retiring vehicles and reflect more realistic situation we have used the vehicle survival functions to estimate the number of vehicles in service. We have reframed the sentence on page 5 line 8 as:

*Moreover, the rapid urbanization has led to an increase in vehicle numbers by about 15-fold over the last two decades, unfortunately increasing the demand for petroleum fuels (DoTM, 2016).*

7)  Line 16: Should this be 'IPCC Tier 1' or 'EMEP/EEA Tier 1' or both?

Response: On page 5 line 19: TIER-I replaced with 'IPCC Tier 1' or 'EMEP/EEA Tier 1'

8) Kerosene lamps. Little distinction is made between kerosene wick lamps and kerosene hurricane lamps – (although 'kerosene lanterns' are referred to once on page 22, line 9, which I assume equates to hurricane lamps?). In Table S2, for BC, only the emission factor (EF) for wick lamps (90 g/kg) is given, from Lam et al (2012), although that paper (Table S5) estimates 20% of kerosene used for lighting in this region is likely to be in hurricane lamps (EF for BC is 9 g/kg). Do the authors therefore assume no hurricane lamp use for their calculations? Also, in Table S2, Lam et al (2012) is given as the source of the N2O and SO2 EFs, but these do not exist in that paper (as far as I can see) and the OC EF of 0.52 g/kg in Table S2 compares with the average of 0.4 g/kg for wick lamps given in Lam et al. Please could the authors correct and/or explain the derivation of their EFs for kerosene-fuelled lighting.

Response: The assumption about kerosene lamps is revised here. Due to a lack of studies on the definite number of types of lamps, it is assumed that 50 % of the population relies on wick lamps and rest on kerosene lanterns. Therefore, a sample weighted average of the kerosene wick and hurricane lamp emission factors are considered (wherever possible) and revised accordingly:

*Lam et al., (2012)*
$CO_2$ – (7*2770 + 3*3080)/(7+3) = 2863 g/kg
CO – (7*11 + 3*3)/(7+3) = 8.6 g/kg
BC – (7*90+3*9)/(7+3) = 65.70 g/kg
OC – (7*0.4+3*0.5)/(7+3) = 0.43 g/kg
$PM_{2.5}$ – (7*93+3*13)/(7+3) = 69 g/kg

*Zhang et al., (2000)*
$SO_2$ – (0.033+0.011)/2 = 0.022 g/kg

*Smith et al., (2000)*
NMVOC – (14.86+19.2)/2 = 17.03 g/kg
$CH_4$ – (0.288+1.071)/2 = 0.68 g/kg
$N_2O$ – (0.079+0.102)/2 = 0.091 g/kg

9) Brick kilns: Page 13, line 2: It would be nice to know how many of the 557 FCBTKs had the zig-zag firing technology, and also how many VSBK there were.

Response: Following sentence added on page 10 line 28:

*The zig-zag firing technique is a relatively new development in Nepal. It has only been used in the brick kilns in the Kathmandu Valley, which were rebuilt after 2015 Earthquake in Nepal. Kilns outside the Kathmandu Valley are slowly adopting the zig-zag technique. An in-house survey of 82 brick factories in 2014 in the Kathmandu Valley showed that only 22 FCBTK (~25%) had zig-zag firing compared to straight ones. This fraction may be extrapolated at a national level to understand the number of zig-zag firing brick kilns in Nepal. A thorough study is indeed required to furnish the actual numbers.*

10) Technical corrections Page 2, line 14: 'Tons' is not and SI unit – presumably this should be 'tonnes? Then Gg is used thereafter. Consistency in use of units required – suggest using Gg throughout (or Tg for CO2 and CO).

Response: Units throughout the manuscript are made consistent. Fuel quantities are specified in tonnes/million tonnes and all emissions are reported in Tg and Gg.

11) Page 4, line 21: Should 'intake' be 'exposure'? Line 22: children (not childrens')

Response: Suggested changes incorporated on page 4 line 21 and line 22.

12) Page 6, line 6: 'arising' not 'arousing'. Line 16, Delete '-forcing' as SLCPs stand for 'short-lived climate pollutants'.

Response: Suggested changes incorporated on page 6 line 11 and line 28.

13) Page 9, line 7: Replace 'rest' with 'the remaining'. Line 8, 'LPI' and 'SMI' should have been defined earlier in this paragraph. Line 26: Insert 'data on' between 'provided' and 'how'.

Response: Suggested changes incorporated on page 9 line 1.

14) Page 11, line 22: Should be 'we intend' not 'we tend'.

Response: Sentence about aviation which had 'we tend' was deleted.

15) Page 13, line 19: Who is the personal (not personnel) communication from?

Response: By personal communication, we refer to a brief study by Dr. Prakash Bhave, one of the co-authors, who along with a Master's student from a local university, as a part of the thesis performed a random check on several vehicles' exhaust before and after servicing the vehicles in the Kathmandu Valley. One of the key findings highlighted the higher percentage of the super-emitters in Nepal (rather than the assumed 20%) due to poor maintenance of the vehicle and road infrastructure.

16) Page 17, line 7: Replace 'they' with 'there'.

Response: Sentence has been modified on page 14 line 26.

17) Page 19, line 2: Replace 'small increase in' with 'slightly higher level of'. Line 6, Replace 'increase' with 'difference'. Line 8, insert 'being' at start of line.

Response: Suggested changes are incorporated on page 16 line 22, line 28 and page 17 line 1.

18) Page 21, line 18: Replace 'spike in 2016 energy' with 'large increase in 2016 energy use'.

Response: Suggested changes are incorporated on page 19 line 11.

19) Page 24, line 1: For 'aerosols' the %SO2 is included in the list yet NOx is also an important precursor of secondary inorganic aerosol – why not include this too? If NH3 were to be added to the inventory (see general comments), this would also need to be added for the same reason.

Response: The NOx emissions are classified under ozone precursors since $NO_X$ plays a vital role in ozone chemistry, like VOCs and CO, and hence is explained in the succeeding paragraph page 23 line 4, although $NH_3$ and $NO_X$ are associated with secondary inorganic aerosols as well.

20) Page 24, line 2: High OC emissions are referred to as shown in Fig 6 – but CO is in that fig, not OC. Has there been a mix-up over OC versus CO?

Response: Although this may at first be confusing, there is not a mix-up between OC and CO. Rather, Fig 6, cited on page 22 line 15 highlights the top six combustion technologies for $PM_{2.5}$, BC, $NO_X$ and CO emissions. Since the top six combustion technologies for OC closely resemble those for $PM_{2.5}$, the additional sub-plot for OC is not shown here. Instead the emphasis is laid on the other distinct sources. A figure showing OC, $SO_2$, NMVOC, $CH_4$, $CO_2$ and $N_2O$ is appended in the supplementary information as Fig S9.

21) Page 22, line 19: The text includes the 2011 emission estimate for CO and then on page 23, lines 5 & 6, emissions of CO2, CH4 and N2O. Why were these not included in Figure 5(a) or perhaps in a separate table?

Response: CO, $CO_2$, $CH_4$ and $N_2O$ emissions are now included in Figure 5(a).

22) Page 25, lines 19 & 20: Values given here for all species apart from SO2 are slightly different from those given in Table 5.

Response: Table 5 was checked carefully again and the values match with the ratio mentioned on page 24 line 2.

23) Page 27, lines 23& 24: Need to rephrase this – I don't think diesel gen-sets were changed to zig-zag firing!

Response: Thank you for catching this. We have rephrased the sentence as follows on page 28 line 13:

*There was a major change in two main polluting sources during 2016, (i) straight firing brick kilns in the Kathmandu Valley were changed to the zig-zag firing technology when they were rebuilt after the 2015 earthquake and (ii) diesel generator sets were phased out due to improvements in load shedding hours since 2016.*

24) Figure 9: Make clear this graph is for 2011.

Response: 'Monthly distribution for 2011' is now mentioned for Figure 9

25) Table 6: Make clear that these values are emission ratios (KTM/NPL).

Response: Emission ratios are now mentioned in the heading of the table:

*Table 4. Comparison of total and sectoral emission ratios for the Kathmandu Valley and Nepal for 2011.*

26) Table S3: Footnote 'e' refers to the liquid fuel combustion in industry having a 22.5% sulfur retention. But liquid fuels leave no ash and so there should be zero sulfur retention in ash – so this must be wrong. If this footnote should have applied to coal use in industry, then again 22.5% looks wrong as USEPA's AP42 (5th edition, Section 1.1.3.2) implies only 5% retention-in-ash for bituminous coal (the type of coal used in Nepal). Please explain.

Response: The Footnote 'e' in Table S2 has been simplified and corrected. Liquid and gaseous fuels had no sulfur retention in the ash. It is now clearly mentioned in the footnote.

*[e]SO2 based on the sulfur content of the fuel. For liquid and gases fuels, there is no sulfur retention in the ash.*

For bituminous coal, 5% retention is assumed as noted in the USEPA's AP42 (5th edition, Table 1.1-3), similarly corrected in Table S2.

*[a]Coal Combustion emission factors, from the USEPA's AP42, Table 1.1-3, Table 1.1-11, Table 1.1-19. SO2 is based on the sulfur content of the fuel. For coal, 5% sulfur retention is assumed from the AP42 document, Table 1.1-3*

---

## Author Response (AR2)

**RESPONSE TO REVIEWERS' COMMENTS**

**Title: "Nepal Emission Inventory (NEEMI): A high resolution technology-based bottom-up emissions inventory for Nepal 2001-2016"**

Pankaj Sadavarte*[a,c], Maheswar Rupakheti*[a], Prakash V. Bhave[b], Kiran Shakya[b], Mark G. Lawrence[a]

[a]Institute for Advanced Sustainability Studies (IASS), Berliner Str. 130, 14467 Potsdam, Germany
[b]International Centre for Integrated Mountain Development (ICIMOD), Lalitpur, Nepal
[c]Now at SRON Netherlands Institute for Space Research, Utrecht, The Netherlands

*Corresponding Authors: Pankaj Sadavarte (p.sadavarte@sron.nl) and Maheswar Rupakheti (Maheswar.rupakheti@iass-potsdam.de)

We thank both referees for their constructive comments. The scientific and editorial comments have been considered and carefully addressed through the changes detailed below. The comments from both referees have been broken down into points and subsequently answered. A point by point response is included indicating where changes appear in the revised manuscript along with the subsequent changes. The referee's comments appear in black and response to the comments follows immediately in blue color.

**Note:**
1. All the changes in the revised manuscript are highlighted in blue color.
2. Line numbers indicated in this text refer to the revised manuscript.

**ANONYMOUS REFEREE #1**

1) Page 11, line 21: Named person missing from 'personal communication' reference.
Response: Personal communication reference name, Prakash Bhave is mentioned on page 11 line 21.

2) Page 22, line 12: Like SO2, NOx is an important precursor of secondary inorganic aerosol. So if SO2 is in this list, why not NOx? Of course, NOx is also a precursor of O3 but then PM2.5 kills more people than O3. Alternatively, replace the word 'aerosols' with 'primary particulate matter' and delete the gas SO2 from this list. Perhaps a minor point - Editor to decide.
Response: The word 'aerosols' is replaced with 'primary particulate matter' on page 22 line 14. $SO_2$ gas is removed from the list and analysis on $SO_2$ has been shifted to next paragraph on page 23 line 14.

3) Page 22, lines 13-15: Issue still not addressed. If OC is not included in Fig 6 then this sentence should not imply that it is.
Response: Since OC is not included in Fig 6, description about OC emissions has been removed from line 15. Additionally a sentence is added on line 17 to describe OC emissions separately.

4) Table S1: Footnote for Kerosene lamps should say 'Assumed 50% kerosene wick lamps..........' i.e the word 'wick' is missing.

Response: Suggested changes are made in supplementary information Table S1 footnote. The word 'wick' is added in the footnote of Table S1 for kerosene lamps.

**RESPONSE TO REVIEWERS' COMMENTS**

**Title: "Nepal Emission Inventory (NEEMI): A high resolution technology-based bottom-up emissions inventory for Nepal 2001-2016"**

Pankaj Sadavarte*[a,c], Maheswar Rupakheti*[a], Prakash V. Bhave[b], Kiran Shakya[b], Mark G. Lawrence[a]

[a]Institute for Advanced Sustainability Studies (IASS), Berliner Str. 130, 14467 Potsdam, Germany
[b]International Centre for Integrated Mountain Development (ICIMOD), Lalitpur, Nepal
[c]Now at SRON Netherlands Institute for Space Research, Utrecht, The Netherlands

*Corresponding Authors: Pankaj Sadavarte (p.sadavarte@sron.nl) and Maheswar Rupakheti (Maheswar.rupakheti@iass-potsdam.de)

We thank both referees for their constructive comments. The scientific and editorial comments have been considered and carefully addressed through the changes detailed below. The comments from both referees have been broken down into points and subsequently answered. A point by point response is included indicating where changes appear in the revised manuscript along with the subsequent changes. The referee's comments appear in black and response to the comments follows immediately after in blue color.

**Note:**
1. All the changes in the revised manuscript are highlighted in blue color.
2. Line numbers indicated in this text refer to the revised manuscript.

**ANONYMOUS REFEREE #2**

I principally welcome the decision about developing two papers and clearly defining the scope of Part I and II. Personally, I'd probably choose a strategy to split the parts by addressing different pollutants since including open burning of agricultural residue and soil NOx would allow to have a complete national inventory of anthropogenic emissions for PM2.5, BC, OC, SO2, NOx, CO, and CO2 while CH4, N2O, NMVOC with large share of fugitive emissions, agriculture (possibly extended with biogenic emissions) would create Part II but also have complete coverage of sources (the combustion bits are done already and could be covered there). The advantage of that would be that for key air pollutants part I would give a complete picture of anthropogenic emissions and the whole discussion of shares/importance of particular sources that is now included in the paper could be compared easily to other work where total national emissions are presented, etc.

Still, the most important thing is to be clear about the coverage of sources and if the authors decide to go with such allocation of sources between part I and II, then I would suggest that Part II includes somewhere a summary of both Part I and Part II, i.e, national totals.

Response: We thank reviewer for constructive suggestion on strategically splitting the parts. We decide to keep the distribution of sectors in part I and II as mentioned in the manuscript. We also accept the comment to include a summary on the anthropogenic

emissions and the national totals emissions in Part-II, which will also include estimates from Part-I as well, making the emission inventory for Nepal more or less complete.

I'd like to thank the authors for responding to all the comments and considering several of them directly in the new revised version of the paper.

For most, I find the newly manuscript well written and documented, in fact some of the parts are a bit long giving account of many details which could be moved to SI. Abstract needs further work and shortening, I think.

Response: We appreciate reviewers perspective on shortening some section and moving them to SI, however since, it's one of a kind work done over Nepal that provides technical details necessary for giving reader proper orientation with brief overview of the situation and for the local and technical interest. Therefore, we would like to keep the manuscript complete in all due respect.

Here are few more specific comments:

1) TITLE: I would suggest to shorten the title a bit and remove the "technology-based" as this is the element that can be included in the keywords and so it could be: "Nepal Emission Inventory – I: A high resolution bottom up emission inventory of combustion sources (NEEMI-Tech) for 2001-2016" or even shorter "A high resolution Nepal emission inventory for 2001-2016: Part I - combustion sources"

Response: We totally agree with the reviewer and would like to shorten the title of manuscript. The title has been changed, pending approval by the editor. We propose it as:

*"Nepal Emissions Inventory – I: Technologies and combustion sources (NEEMI-Tech) for 2001-2016"*

The part II will have the following title:
*"Nepal Emissions Inventory – II: Open burning and fugitive sources (NEEMI-Open) for 2001-2016"*

2) ABSTRACT: I think the current abstract is too long and contains too many details. I think it shall address only key and not elements of the paper highlighting key results and possibly knowledge gaps. At the same time, it would be important to add one statement earlier in the abstract (e.g., around the 2nd sentence) that clearly defines the scope saying it is part I covering this and that. Part of the sentences like in line 5 after approach can be deleted, especially that I do not think the inventory is done for the purpose of understanding technologies or sectoral energy consumption. Also several statements in parenthesis (like the ones about machinery, non-renewability of biomass) can be also deleted. Suggest to simplify the part describing key sources from line 15 onwards. Somehow it is hard to see right away what are the key sources.

Response: Abstract is trimmed down and rewritten to highlight key results of the paper. A sentence citing the second part of the complete study is now mentioned on line 5 of the

abstract. The part of the sentence in line 5 "the purpose of understanding technologies or sectoral energy consumption" is now deleted. Several short statements like the ones about machinery, non-renewability of biomass in parenthesis are deleted.

The new abstract now reads as follows:

The lack of a comprehensive, up-to-date emissions inventory for the Himalayan region is a major challenge to understanding the extensive regional air pollution, including its causes, impacts, and mitigation pathways. This study describes a high resolution (1 km × 1 km) present-day emission inventory for Nepal, developed with a higher-tier approach. The complete study is divided into two parts; this paper covers technologies and combustion sources in residential, industry, commercial, agricultural diesel-use and transport sectors as part-I (NEEMI-Tech), while emissions from open burning of municipal waste and agricultural residue in fields, and fugitive emissions from waste management, paddy fields, enteric fermentation and manure management for the period 2001–2016 will be covered in part-II (NEEMI-Open). The national total energy consumption estimated in the base year 2011 was 374 PJ, with the residential sector being the largest energy consumer (79 %) followed by industry (11 %) and the transport sector (7 %). Biomass is the dominant energy source, contributing 88 % to the national total energy consumption, while the rest is from fossil fuel. A total of 8.9 Tg $CO_2$, 110 Gg $CH_4$, 2.1 Gg $N_2O$, 64 Gg $NO_X$, 1714 Gg CO, 407 Gg NMVOC, 195 Gg $PM_{2.5}$, 23 Gg BC, 83 Gg OC and 24 Gg $SO_2$ emissions were estimated in 2011 from the five energy-use sectors considered in NEEMI-Tech. NEEMI provides for the first time temporal trends of fuel and energy consumption and associated emissions in Nepal for a long period, 2001-2016. The energy consumption showed an increase by a factor of 1.6 in 2016 compared to 2001, while the emissions of various species increased by a factor of 1.2–2.4. An assessment of the top polluting technologies shows particularly high emissions from traditional cookstoves and space heating practices using biomass. In addition, high emissions were also computed from fixed chimney Bull's Trench kilns in brick production, cement kilns, two-wheeler gasoline vehicles, heavy diesel freight vehicles and kerosene lamps. The monthly analysis shows December, January and February as periods of high $PM_{2.5}$ emissions from the technical sources considered in this study. Once the full inventory including open burning and fugitive sources (part-II) is available, a more complete picture of the strength and temporal variability of the emissions and sources will be possible. Furthermore, the large spatial variation in the emissions highlights the pockets of growing urbanization, which emphasizes the importance of the detailed knowledge about the emission sources that this study provides. These emissions will be of value for further studies, especially air quality modelling studies focused on understanding the likely effectiveness of air pollution mitigation measures in Nepal.

3) Page 10: line 2; reference to Baidaya and Borkenkleefeld…the proper name of the second author is Borken-Kleefeld

Response: We thank reviewer for bringing to our notice the following error. It's been corrected on page 10 line 2.

4) Page 10: line 22-24; here different technologies are listed and also info is given about number of different kilns. Table 1, however, lists only FCBTK. I think it might be reasonable to add all of them to table 1.

Response: We thank reviewer for pointing this out. Table 2 has been updated. Now it includes the list of combustion technologies and therefore technologies under bricks manufacturing is updated to include all the known combustion technologies like FCBTK-straight firing, zig zag firing, clamps and vertical shaft brick kilns (VSBK).

5) Page 18, line 22-26 and section 3.2 as well as other sections where reference to 'national' emissions are made; I think it would be useful to repeat few times that the presented shares refer to the combustion emissions only and not the total – just a reminder to the reader that these are not necessarily national totals.

Response: Following sentences are added to clarify the national estimates.

Page 19 line 20: *The emissions discussed henceforth refer to the estimates from the sectors and source categories described above, and do not account for the complete national totals, which will also include emissions from Part-II of this work.*

Page 20 line 21: *... from the sectors discussed in this part of the work. However the total national values will also include emissions from the second part of the study (NEEMI-Open).*

We have also replaced 'national' word in the context of emissions, with the apt information that explains the extent of emissions covered in this part of the study. Here are few changes mentioned with page and line number:

*'nationally' replaced with 'From the total of five sectors' on*
*Page 20 line 4*

*'national' replaced with 'five sectors' on*
*Page 20 line 26; Page 23 line 9 and Page 25 line 17*

*'Nationally' replaced with 'A total' on*
*Page 2 line 13;  Page 21 line 9  and   Page 30 line 27*

*'...national emissions estimates of aerosols, ozone precursors and greenhouse gases' replaced with '...emissions estimates of aerosols, ozone precursors and greenhouse gases from the five energy-use sectors on Page 22 line 13*

*'...of the national estimate' replaced with '...from sources in this part of the study' on*
*Page 23 line 6*

*'At the national level' replaced with 'For the five sectors considered' on*
*Page 26 line 3*

*'national' replaced with 'total' on*
*Page 23 line 21;*
*Page 24 line 6;*
*Page 25 line 10;*
*Page 26 line 12*
*Page 26 line 27;*
*Page 27 line 18;*
*Page 28 line 5;*
*Page 31 line 17*

6) Section 3.4.1: In a number of places the authors refer to the argument that some differneces are there due to the difference or assumption about the sulfur retention in ash. It is not clear to me if this refers to the NEPAL inventory or the inventories which are compared like CMIP6 or EDGAR or GAINS-ECLIPSE?

Response: We thank reviewer for highlighting the confusion in the analysis. We have now reframed the sentences which now clarify the $SO_2$ emission factors from NEEMI on page 24 line 20.

*"Also, the CMIP6 SO2 emissions from each sector vary a lot when compared to NEEMI, showcasing the shortcomings due to the coarser resolution methodologies, that lack the degree of detail present in the NEEMI inventory. Like in NEEMI, SO2 emission factors used are for a large number of technology-fuel combinations, the sulfur content of the liquid fuels changes over a period of one and half decades and the sulfur retention fraction in the ash content of coal depends on the combustion technology."*

7) Page 23: line 15; I think the authors need to be more precise when they refer to the "non-renewability factor" they should explain the term, its relevance and the assumption made with respect how much of the biomass is considered non-renewable and how it affects CO2 calculation presented in the paper.

Response: We have now mentioned the importance of non-renewability factors and the details about this in the supplementary information, rather than the main manuscript, to avoid redundancy and diluting the main analysis.

Non-renewability fraction for biomass (NRB) can be defined as the imbalance between demand and supply which contributes to net-$CO_2$ emissions. Ghilardi et al., (2007) explains it as *"when the amount extracted and burned exceeds the growth rate of the living biomass sources"*, it contributes to net-$CO_2$ emissions. In simple terms, Venkataraman et al., (2010) defines NRB as *"the percent of woodfuel that is harvested on a non-renewable basis"*. The NRB factor plays a crucial role in estimating CO2 or carbon budgets. The net-$CO_2$ emissions from harvested fuelwood or biomass products can help identify actual carbon offsets achieved through Clean Development Mechanism (CDM) projects, although it's not the only criteria. The NRB fraction used over Nepal is 10% similar to Venkataraman et al., (2010) over India based on residential sector fuelwood supply and demand. In Nepal, residential sector consumes 79 % of the national energy during 2011 and biomass is the single largest source of energy attributing to 88 %

of the national energy. Therefore, we have used 10 % NRB in our study to calculate $CO_2$ emissions. In principle the % NRB is calculated as *% NRB = (fuelwood demand – fuelwood supply)/fuelwood supply* (Venkataraman et al., 2010, Ghilardi et al., 2007, 2009). Studies like Ghilardi et al., (2009) have estimated non-renewable fuelwood fraction ranging from 0 to 96 % based on demand and supply at local level in Central Mexico using GIS method. These NRB fractions can then be used to allocate the percentage of fuelwood that can be treated as non-renewable fuel and their emissions can be accounted as the net-$CO_2$ estimate. In our study, we have considered for 10 % NRB which means out of total fuelwood consumed, combustion of 10 % would give net-$CO_2$ emissions while the rest 90 % can be treated as sustainable fuelwood and doesn't contribute directly to the net-$CO_2$ emissions.

8) Page 25: line 12-14: As I mentioned earlier, the CMIP6 and NEI inventory might differ in source coverage and for example her the NOx from soils (agriculture) could be one of these differences. I think this can be eliminated (and shall be) for the comparison as the CEDS/CMIP6 inventory data is explicit about this particular source and so it can be subtracted for comparison.

Response: We have deleted the sentence.

9) Page 46: Figure 7 and respective text in the paper; It might be useful to add in the text a word or two about potential issues of not full compatibility of sectoral structures of different inventories and the NEPAL inventory, i.e., a source of uncertainty in the comparisons that will be different from pollutant to pollutant but in a way is unavoidable. Additionally, I am not sure if the authors used the gridded data sets to estimate the numbers from global inventories or reached out to the authors or the web sites where respective sectoral data is available. I think it would be good to say explicitly and maybe even add the web links.

Response: We have added the link to emission sources for MIX, REAS 2.1, EDGARv4.3.2, CMIP6 and ECLIPSE V5a-GAINS in parenthesis that were used for comparison with NEEMI inventory:

- MIX (emission source: Nepal emissions reported in Li et al., 2017) on page 23 line 2
- REAS 2.1 (emission source: https://www.nies.go.jp/REAS/) on page 23 line 11
- EDGARv4.3.2 (emission source: https://edgar.jrc.ec.europa.eu/overview.php?v=432_GHG&SECURE=123; https://edgar.jrc.ec.europa.eu/overview.php?v=432_AP) on page 24 line 1
- CMIP6 (emission source: https://www.geosci-model-dev.net/11/369/2018/gmd-11-369-2018-assets.html) on page 24 line 3
- ECLIPSE V5a-GAINS (emission source: Author/co-author reachout) on page 24 line 4